# Sub-femtomolar drug monitoring via co-calibration mechanism with nanoconfined DNA probes

Yonghuan Chen [1,2], Xiuying Li[1], Xinru Yue[1], Weihua Yu[1], Yuesen Shi[3], Zilong He[1], Yuanfeng Wang[4], Yu Huang [2] ✉, Fan Xia [2] & Fengyu Li [1,5] ✉

Synthetic drugs fundamentally reshape the illicit drug market due to their low cost, ease of production, and rapid manufacturing processes. However, current drug detection methods, which rely on complex instruments, have limited applicability and often neglect the influence of pH fluctuations, leading to potential bias and unreliable results. Herein, we propose co-calibration DNA probes on a nanoconfined biosensor ($N_CB_S$), covering the range of sweat pH 3–8 to achieve significantly enhanced target signal recognition. The $N_CB_S$ exhibits a linear response range of $10^3$-$10^8$ fM with a low limit of detection (LOD) of 3.58 fM in artificial sweat. Compared to the single-aptamer $N_CB_S$, the dual-aptamer $N_CB_S$ offers a broader linear response range, primarily due to the synergistic effects of changes in surface wettability and the capture of hydrion, which together reduce signal interference in proton transport. The linear response range doubles, and its detection sensitivity improves by 4–5 orders of magnitude compared to existing drug detection methods. This sensing strategy expands the application scope of aptamer-based composite probes, offering an approach for ultra-sensitive drug detection and demonstrating significant potential in sweat sensing and drug monitoring fields.

The 2023 World Drug Report indicated that the global number of drug users continues to rise, with over 296 million people using drugs worldwide in 2021, a 23% increase compared to a decade ago[1]. The ongoing expansion of the illegal drug market and the increasing flexibility of drug trafficking networks pose severe challenges to law enforcement and healthcare systems[2–4]. The ultra-sensitive detection and application of drugs can effectively prevent and curb crime. Traditional drug detection methods include gas chromatography-mass spectrometry (GC-MS), liquid chromatography-mass spectrometry (LC-MS), and commercial enzyme-linked immunosorbent assays (ELISA)[5–9]. These detection methods primarily rely on large-scale instruments, which, despite their high sensitivity, are unlikely to be applicable for on-site testing. Recently, sweat samples have become ideal for real-time detection using portable devices because they are easy to collect and do not require special storage conditions[10,11]. In particular, on-site drug monitoring in sweat, compared to blood and urine testing, is both non-invasive and does not involve privacy concerns[12–14]. However, pH variations in sweat can introduce interference, affecting detection performance. Therefore, there is an urgent need to develop sensor-based ultra-sensitive detection technologies for on-site drug testing in sweat.

[1]College of Chemistry and Materials Science, Guangdong Provincial Key Laboratory of Speed Capability Research, Su Bingtian Center for Speed Research and Training, Jinan University, Guangzhou, PR China. [2]State Key Laboratory of Biogeology and Environmental Geology, Engineering Research Center of Nano-Geomaterials of Ministry of Education, Faculty of Material Science and Chemistry, China University of Geosciences, Wuhan, PR China. [3]Anti-Drug Technology Center of Guangdong Province, Guangdong Province Key Laboratory of Psychoactive Substances Monitoring and Safety, Guangzhou, PR China. [4]Key Laboratory of Evidence Science, China University of Political Science and Law, Beijing, PR China. [5]College of Chemistry, Zhengzhou University, Zhengzhou, PR China. ✉e-mail: yuhuang@cug.edu.cn; lifengyu@jnu.edu.cn

In recent years, solid-state nanochannels have shown broad application potential in various technological fields due to their precise molecular-level control and tunable structural properties[15–20]. Compared to other detection approaches, nanoconfined biosensors ($N_CB_S$) endow significant advantage on high sensitivity[21,22]. By integrating resistive pulse sensing technology and utilizing functional molecules (such as crown ethers[23–26], DNA probes[27–29], etc.), specific functionalities can be imparted to nanochannels, enabling regulation of effective pore size, surface wettability, and surface charge properties. When a target analyte binds to specific recognition sites on the $N_CB_S$, the ion transport characteristics of the electrolyte ions passing through the nanochannel under an applied voltage change, thus enabling the detection of the target analyte[30–33]. DNA probes, single-stranded DNA or RNA oligonucleotide sequences can be obtained through an in vitro selection process known as Systematic Evolution of Ligands by Exponential Enrichment[34]. This endows them with high stability, strong affinity, and significant specificity for target molecules. DNA-$N_CB_S$ holds great significance for the detection of ions and small molecules[35–39].

In conventional nanopore, measurements are typically conducted under a single pH condition, overlooking the impact of pH variation on detection performance. However, variations in pH can directly impact the hydration states of ions and their transport behavior within the nanopore[40–42], and even modify the interaction network within the confined space[43]. Consequently, shifts in pH alter the surface charge density of the nanopore walls and the distribution of electric potential, thereby affecting the local arrangement of water molecules and associated hydration state[44,45]. Under these conditions, highly charged nanopores may slow or even reverse protein transport based on pH, further underscoring the critical role of pH in governing selective transport phenomena[46]. Such alterations have profound implications for target recognition and sensitivity, especially when detecting complex biological samples like sweat, where the pH fluctuates between 3.0 and 8.0[47]. This instability reduces the detection performance of conventional nanopores. The DNA-$N_CB_S$ can maintain stable performance across different pH values and significantly enhance the recognition capability for complex samples[48–50]. Therefore, it is essential to develop nanopore devices with co-calibration capabilities for real samples like sweat, which exhibit a wide range of pH variations. Inspired by the roles of AG and SF in biological nanopores[51,52], the co-calibration of different responsive molecules enhances target recognition sensitivity and resistance to external interference, thereby amplifying the recognition signal within the nanopores. This co-calibration DNA-$N_CB_S$ has the potential to be further developed into a wearable sensor by integrating microfluidic technology (Fig. 1a), enabling convenient, rapid, and ultra-sensitive sweat sensing and health diagnostics[53–55].

In this work, we develop a co-calibration strategy with biomimetic solid-state $N_CB_S$ with dual-DNA probes. The $N_CB_S$ utilizes a cathinone-binding aptamer (CBA) as the gate molecule for specific sensing (AG). Upon binding to the target, the CBA transitions from a long-chain structure to a hairpin configuration. Concurrently, C4 DNA serves as the functional molecule (SF) regulating selective ion transport. Its structure responds to pH changes by forming an i-Motif structure under acidic conditions when binding to hydrion (Fig. 1b). This enhanced performance is mainly attributed to the reduced hydrophilicity of the $N_CB_S$ resulting from dual-aptamer functionalization, which amplifies the current change when CBA binds to the target (Fig. 1c, Supplementary Fig. 1). Additionally, C4 DNA captures hydrion from the solution, effectively minimizing signal interference from ion transport. The difference in length between the two DNA probes also increases the density of DNA-$N_CB_S$, leading to a more uniformly distributed positive charge (Supplementary Fig. 2 and Supplementary Table 2). This uniform charge distribution amplifies the significant charge changes upon target binding, thereby improving the target recognition signal. The investigation of drug monitoring in artificial sweat demonstrates that the CBA&C4@AAO biosensor maintains ultra-sensitivity and a broad linear response range, even in complex environments.

## Results

### Design and sensing performance of single DNA probe $N_CB_S$

The design strategy for the solid-state $N_CB_S$ functionalized with single DNA probes CBA@AAO is shown in Supplementary Fig. 3. And the successful preparation of CBA@AAO was demonstrated in detail by scanning electron microscope (SEM), energy dispersive spectrometer (EDS) mapping, X-ray photoelectron spectroscopy (XPS), contact angle and electrochemical tests (Supplementary Figs. 4–14 and Supplementary Tables 3–9). Subsequently, to demonstrate that the ultra-sensitivity of CBA@AAO to the target molecule is due to the specific probe CBA (Supplementary Fig. 15), we used a PolyA DNA sequence with the same number of bases as CBA but consisting entirely of adenine bases (PolyA) and grafted it onto the AAO membranes. We evaluated changes in the transmembrane ionic current (TmIC) signals of CBA@AAO and PolyA@AAO after exposure to 1 fM and 1 pM Cathinone (Cat) (Supplementary Fig. 16). For CBA@AAO, the current increased with Cat concentrations, whereas PolyA@AAO showed no significant change (Fig. 2a). To further examine stability, we measured the TmIC after CBA@AAO recognized varying Cat concentrations. As shown in Fig. 2b, the TmIC rose with increasing Cat levels, primarily due to the enlarged effective nanochannel diameter resulting from probe-target interactions.

We also investigated direct Cat detection by monitoring TmIC changes. In this mode, the current declined once CBA@AAO recognized Cat (Fig. 2c), and the I-T signal exhibited a downward trend with increasing Cat concentration. At high Cat concentrations, as shown in Fig. 2d, during the process of CBA as a specific binding probe transporting Cat, it gradually blocks the nanochannel and shields the negatively charged surface of the nanochannels, leading to varying degrees of pore blockage[56]. Supplementary Fig. 17 shows the eight-cycle stability of direct detection of 1 pM Cat by CBA@AAO. The primary influencing factor for probe-modified $N_CB_S$ is the size effect. CBA@AAO with pore sizes ranging from 40 to 70 nm exhibited the optimal response to Cat, with TmIC change rates of 10.1% at 1 fM and 22.4% at 1 nM concentrations (Fig. 2e). This superior performance can be attributed to the appropriate pore size and porosity, which play crucial roles during the functionalization process. An optimal combination of these factors ensures not only adequate surface area for CBA modification on the $N_CB_S$ but also sufficient probe functionalization within the pores (Supplementary Fig. 18 and Supplementary Table 10).

To emphasize the sensing performance of the CBA@AAO $N_CB_S$ for detecting the target, we examined the relationship between TmIC and Cat concentration. The results demonstrated a strong linear response of CBA@AAO over the Cat concentration range of $1–10^4$ fM ($R^2 = 0.9984$), with a detection limit as low as 0.40 fM (Fig. 2f and Supplementary Figs. 19, 20). This indicates the ultra-sensitivity and effectiveness of the $N_CB_S$ for ultra-trace detection of Cat. Specificity is crucial in evaluating probe sensor performance. We tested the CBA@AAO binding to methcathinone (Met) and ethcathinone (Eth), which share similar structures with Cat. As shown in Fig. 2g, at 1 fM Cat, the current ratio before and after recognition was 1.10, whereas for Met and Eth, the ratios were only 1.03 and 1.01, respectively, at $10^5$ fM. And the ratio increased to 1.23 at $10^5$ fM Cat. We also compared responses to 11 other drugs, including norketamine (Nor), ketamine (Ket), amphetamine (Amp), MDMA, phenacetin (Phe), cocaine (Coc), heroin (Her), caffeine (Caf), procaine (Pro), paracetamol (Par), and (+)-pseudoephedrine (PSE) (Supplementary Figs. 21–23). CBA@AAO showed strong selectivity for Cat, with varying responses to other substances, likely due to differences in structural features (such as the number and position of amino groups, conjugated structures, etc.).

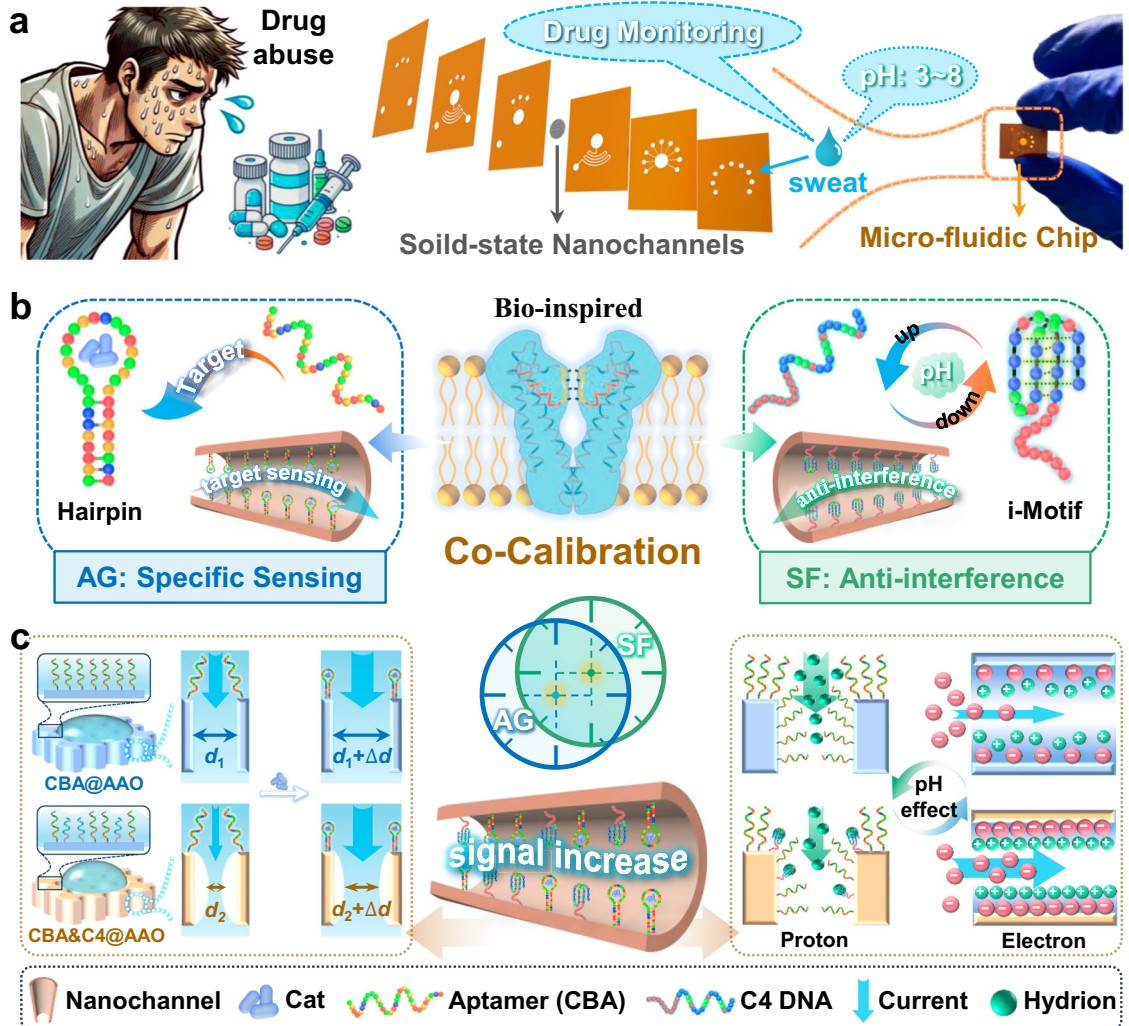

**Fig. 1 | Co-calibration of dual-DNA probes in nanoconfined channel (CBA&C4@AAO) for drug monitoring. a** To address on-site drug detection scenarios, nanoconfined biosensor ($N_CB_S$) can be further developed into a wearable sweat sensor by integrating microfluidic technology, enabling non-invasive, privacy-preserving detection in the future. **b** In biological nanopores, different functional proteins are distributed, including ion-selective proteins (SF) and specific sensing gate proteins (AG). The Cat-specific binding probe (CBA) forms a hairpin structure upon binding to Cat (left diagram). Meanwhile, the pH-responsive C4 DNA undergoes reversible structural changes with varying pH levels (low pH: i-Motif structure, high pH: extended linear structure) (right diagram). **c** The dual-DNA probes in the CBA&C4@AAO provide ultra-sensitive target recognition by altering the effective pore diameter, surface wettability, and other nanochannel properties through probe conformational changes, affecting the ionic current signal. The CBA&C4@AAO surface is functionalized with two distinct DNA probes, resulting in a rougher texture compared to the single-probe CBA@AAO, which slightly reduces hydrophilicity and amplifies the effective pore size variation before and after Cat binding (left diagram). Additionally, the CBA&C4@AAO surface is enriched with C4 DNA, which binds specifically to protons, reducing signal interference caused by proton transmembrane transport (right diagram). The $N_CB_S$ also shows more pronounced surface charge changes, and this synergistic effect significantly enhances the $N_CB_S$ response to Cat detection. This strategy would allow for convenient, rapid, and ultra-sensitive detection of drugs in sweat, as well as applications in health diagnostics.

We successfully differentiated Cat, Met, and Eth using principal component analysis (PCA) based on variations in current signals before and after CBA@AAO response (Supplementary Fig. 24). The distribution of clusters corresponded with increasing concentrations, achieving 94% accuracy (Supplementary Table 11). Building on this, we applied PCA to 14 different drugs at the same concentration, achieving 100% correct classification (Supplementary Fig. 25 and Supplementary Table 12). Additionally, hierarchical cluster analysis (HCA), an unsupervised multivariate method[57,58], was used to distinguish analytes through dimensionality reduction (Fig. 2h). These findings indicate that, beyond specific Cat recognition, the CBA@AAO $N_CB_S$ exhibits cross-reactivity to structurally similar drugs, offering potential for discriminative analysis in complex environments.

## Sensing properties of C4@AAO and CBA&C4@AAO at different pH levels

To address pH interference in Cat recognition, we introduced a pH-responsive specific probe (C4 DNA) and functionalized it on $N_CB_S$, creating the C4@AAO sensor. As shown in Fig. 3a, the C4@AAO exhibited significant variations in *I-V* characteristics in response to different pH levels. At +2 V, the TmIC varied notably: 85.6 ± 1.3 μA at pH 3 (open state) and 20.2 ± 0.1 μA at pH 8 (closed state). The C4-$N_CB_S$ also showed a strong linear response between pH 3-7 ($R^2 = 0.9881$) (Fig. 3b), attributed to pH-induced structural changes in C4 DNA. Cyclic stability tests at pH 5.5 and 7.5 confirmed the C4-$N_CB_S$ robust and reliable performance in pH analysis (Fig. 3c and Supplementary Fig. 26).

To achieve co-calibration for ultra-sensitive Cat detection and pH interference resistance, we developed the CBA&C4@AAO by co-

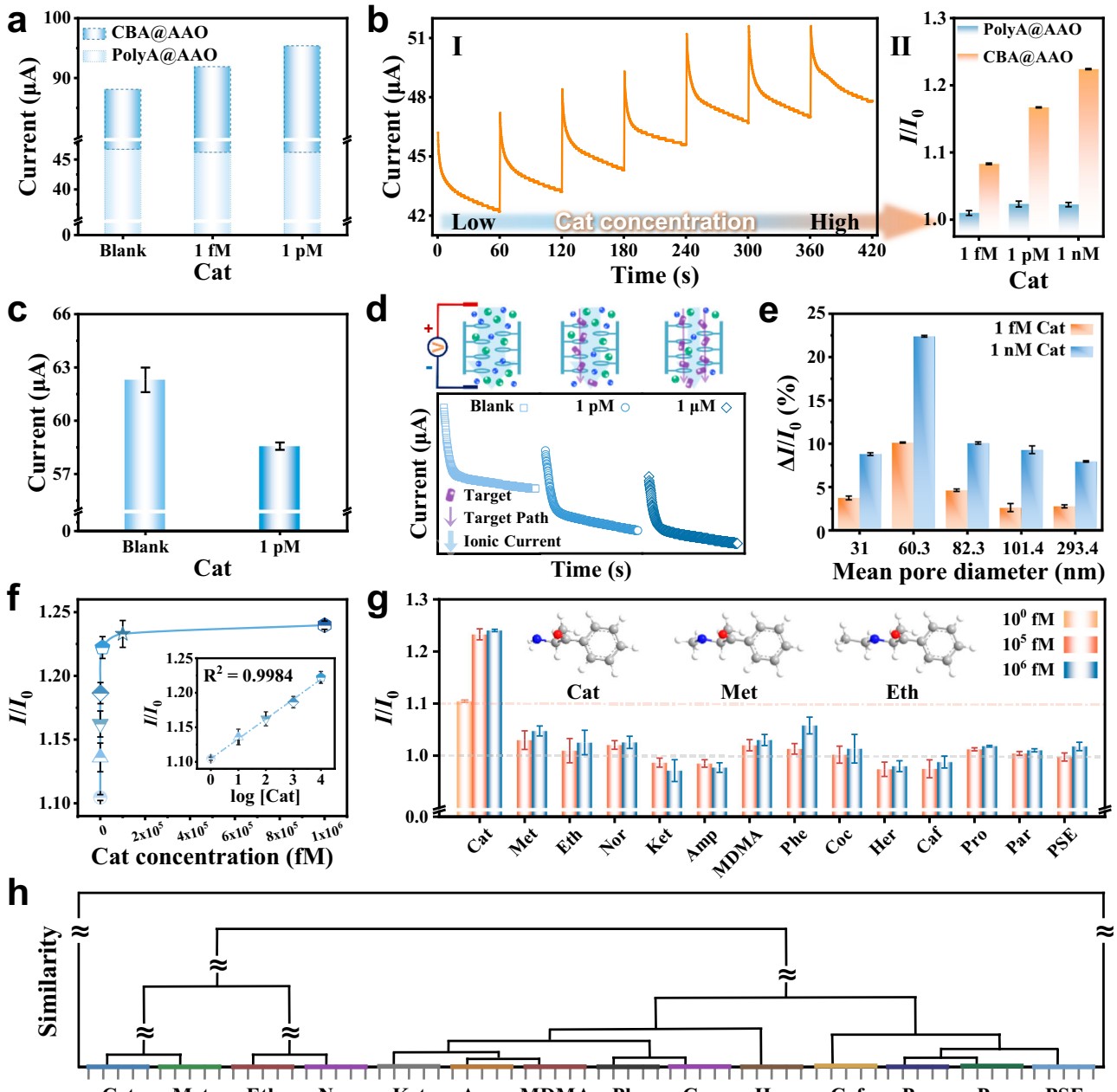

**Fig. 2 | Sensing performance of CBA@AAO $N_CB_S$. a** Stacked column chart of CBA@AAO and PolyA@AAO incubated with different concentrations of Cat, with the current magnitude corresponding to the rapid response current at +2 V in the $I$–$V$ characteristics. **b** $I$–$T$ characteristic graph of the $N_CB_S$ with increasing Cat concentration, with response times of concentration gradients at 60 s (left diagram). The current ratio before and after response to different concentrations of Cat under steady-state conditions (right diagram). **c** CBA@AAO directly detects the current change before and after 1 pM Cat. **d** $I$–$T$ characteristics of CBA@AAO for direct detection of different concentrations of Cat and schematic representation of potential transport mechanisms. **e** Pore size compatibility (size effect) of the probe-$N_CB_S$. **f** Current ratio recorded at different target Cat concentrations based on the $I$–$V$ characteristics at +2 V. The inset shows the linear response of CBA@AAO to Cat concentrations ranging from 1–$10^4$ fM, with $R^2 = 0.9984$. **g** Selectivity of probe-

$N_CB_S$. Thirteen different drugs were compared based on the current ratio before and after recognition at +2 V in the $I$–$V$ characteristics, with Cat concentrations of 1, $10^5$, and $10^6$ fM, and interference substance concentrations of $10^5$ and $10^6$ fM. The inset shows the structural formulas of Cat, Met, and Eth. (Cat cathinone, Met methcathinone, Eth ethcathinone, Nor norketamine, Ket ketamine, Amp amphetamine, MDMA 3,4-methylenedioxyamphetamine, Phe phenacetin, Coc cocaine, Her heroin, Caf caffeine, Pro procaine, Par paracetamol, PSE (+)-pseudoephedrine). **h** Hierarchical Cluster Analysis (HCA) score plot was used to distinguish 14 drug analytes (all at 1 nM) through CBA@AAO. Data in the bar plots (**b**, **c**, **e**, **g**) and the dot plot (**f**) are presented as mean ± standard deviation values derived from the results of three independent measurements (*N* = 3). The error bars represent standard deviation values.

grafting CBA and C4 DNA (1:1 ratio) onto $N_CB_S$ (Supplementary Figs. 27, 28). The biosensor is capable of simultaneously sensing the target and pH changes. We investigated the TmIC changes of CBA@AAO and CBA&C4@AAO before and after target binding under different pH conditions (Fig. 3d). The current measurements were

based on $I$–$V$ characteristics, reflecting the TmIC changes before and after Cat recognition at +2 V. The results showed that the incorporation of C4 DNA led to a significant increase in the current variation upon target recognition, compared to CBA@AAO. As shown in Fig. 3e, at pH 5.5 and 7.5, the current changes of CBA&C4@AAO after

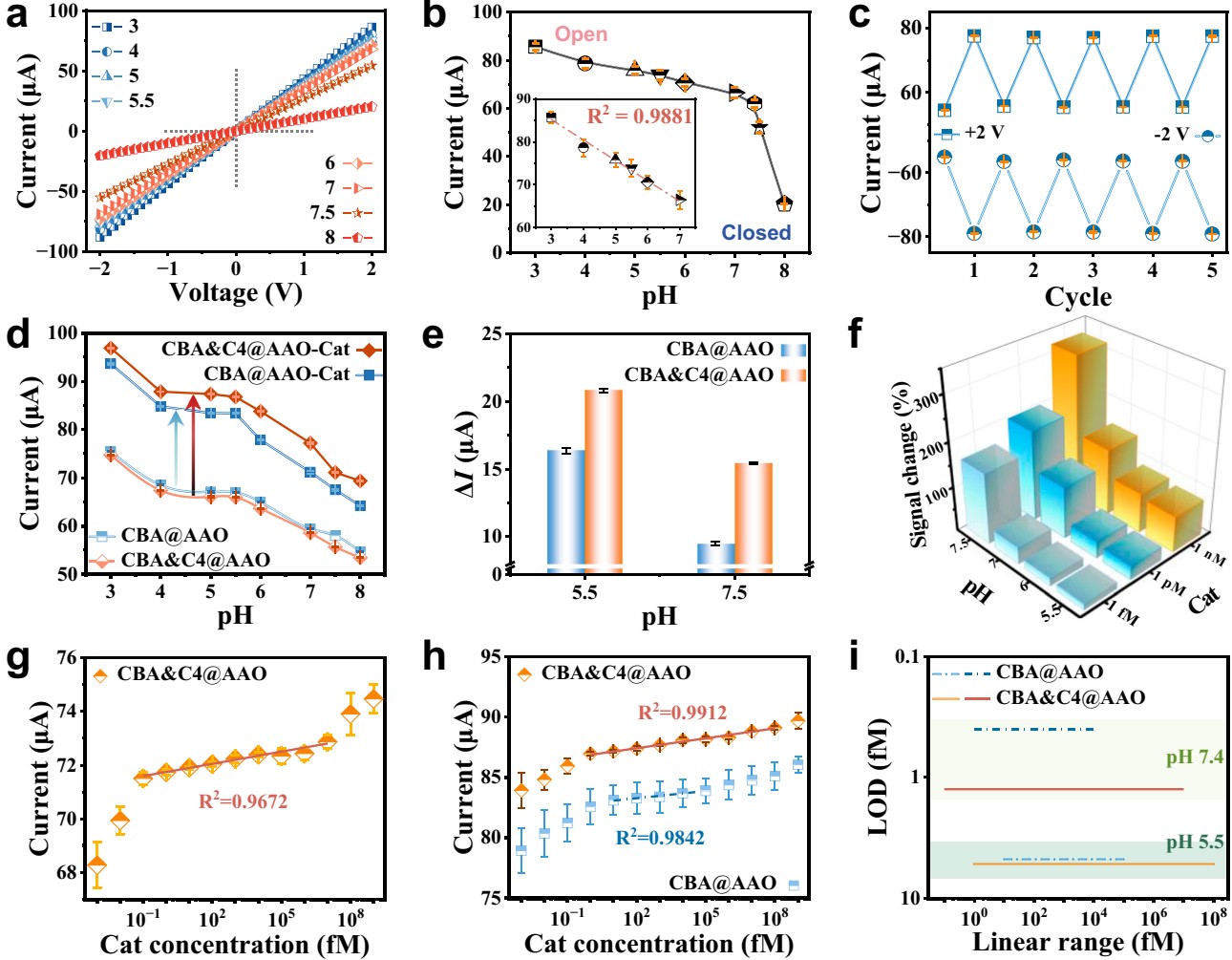

**Fig. 3 | Sensing properties of C4@AAO and CBA&C4@AAO at different pH levels. a** *I-V* characteristics of the C4@AAO in response to different pH levels. **b** Current values of the C4@AAO at +2 V in response to different pH levels. The inset shows the linear fit of the $N_CB_S$ response within the pH range of 3-7, with an $R^2$ value of 0.9881. **c** Cycle stability response of C4@AAO at applied voltages of ±2 V under pH conditions of 5.5 and 7.5. **d** Based on the *I-V* characteristics, the ionic current responses of CBA@AAO and CBA&C4@AAO to 1 nM Cat were measured under pH conditions ranging from 3 to 8 at +2 V. **e** The current changes at +2 V for both CBA@AAO and CBA&C4@AAO before and after Cat binding were further examined at pH 5.5 and 7.5. **f** The current signal responses of CBA&C4@AAO to different concentrations of Cat under varying pH conditions, relative to the signal changes caused by pH interference. **g** The response of CBA&C4@AAO to varying concentrations of Cat at pH 7.4. **h** At pH 5.5, the *I-V* current responses (+2 V) of CBA@AAO and CBA&C4@AAO to different Cat concentrations. **i** The linear response range and a low limit of detection (LOD) of CBA@AAO and CBA&C4@AAO at two different pH levels. At pH 7.4, CBA@AAO demonstrates a linear response range of 1–$10^4$ fM, with LOD of 0.40 fM, whereas CBA&C4@AAO shows a linear range of 0.1–$10^7$ fM and LOD of 1.25 fM. At pH 5.5, CBA@AAO has a linear response range of 10–$10^5$ fM with LOD of 4.78 fM, while CBA&C4@AAO achieves a linear response range of 1–$10^8$ fM and LOD of 5.24 fM. In the co-calibration strategy based on DNA probes regulation of $N_CB_S$ sensitivity, the linear response range of the target can be significantly expanded, despite a slight decrease in the detection limit. Data in the dot plots (**b**, **c**, **d**, **g**, **h**) and the bar plots (**e**, **f**) (Supplementary Fig. 29) are presented as mean ± standard deviation values derived from results of three independent measurements (*N* = 3). The error bars represent standard deviation values.

Cat binding were 20.79 μA and 15.42 μA, respectively, while for CBA@AAO, the changes were only 16.37 μA and 9.48 μA. The CBA provides target specificity, while C4 DNA responds to protons, reducing interference caused by ion transport. By compensating for the pH-induced ionic current changes, the sensitivity of CBA&C4@AAO to Cat recognition was significantly enhanced under varying pH conditions (Fig. 3f and Supplementary Fig. 29). At pH 5.5, the current signal increased by 14%, 31%, and 80% for 1 fM, 1 pM, and 1 nM Cat, respectively. At pH 7.5, the signal increases were even more pronounced, reaching 159%, 201%, and 322%. These findings indicate that the CBA&C4@AAO exhibits excellent potential in resisting pH interference during Cat detection, offering a promising approach to further enhancing $N_CB_S$ sensitivity.

We assessed the linear response ranges and minimum detection limits of two $N_CB_S$ under pH 7.4 and 5.5 levels. As shown in Fig. 3g, the CBA&C4@AAO exhibited a broader linear response range compared to CBA@AAO at pH 7.4. Similarly, at pH 5.5, the TmIC change of CBA&C4@AAO upon binding to Cat was significantly higher than that of CBA@AAO, with a wider linear response range (Fig. 3h). This enhancement is primarily attributed to the introduction of C4 DNA, which slightly reduces the hydrophilicity of the nanochannels membrane, thereby lowering the initial current signal threshold. Additionally, C4 DNA provides numerous hydrion binding sites, mitigating the interference from proton transport. The mixed modification, due to the different lengths of the two DNA strands, increases the probe modification density, resulting in a more uniform

charge distribution on the membrane surface. Consequently, the changes in nanochannels size, surface wettability, and surface charge before and after the binding of Cat in CBA&C4@AAO synergistically enhance the detection signal. The CBA@AAO displayed a linear response range of $1$–$10^4$ fM with a limit of detection (LOD) of 0.40 fM, while CBA&C4@AAO had a linear range of $0.1$–$10^7$ fM and an LOD of 1.25 fM at pH 7.4. At pH 5.5, CBA@AAO exhibited a linear range of $10$–$10^5$ fM with an LOD of 4.78 fM, whereas CBA&C4@AAO showed a linear range of $1$–$10^8$ fM and an LOD of 5.24 fM (Fig. 3i). Although the mixed modification slightly reduced the LOD, it significantly expanded the linear response range, optimizing the co-calibration strategy for enhancing sensitivity based on DNA probes.

### Sensing mechanism of $N_CB_S$ modified by DNA probes

To investigate the reasons for the enhancement of sensitivity and specificity of the probes-$N_CB_S$, we used molecular docking (MD) to predict the binding mode and binding sites of the ligand interacting with the macro-molecule. MD studies can further provide information on the binding affinity between Cat and DNA. As shown in Fig. 4b, the MD results indicate that CBA interacts with Cat through groove binding, primarily through non-covalent interactions (including hydrogen bonds and van der Waals forces), with a binding energy of ~5.6 kcal mol$^{-1}$. This is consistent with the results obtained from circular dichroism (CD) measurements, indicating a conformational change upon binding of CBA to the target Cat (Fig. 4c and Supplementary Fig. 30). With the continuous increase in Cat concentration, the CD signal change becomes more pronounced, accompanied by an increase in corresponding UV absorption (Fig. 4d). Therefore, the proposed mechanism primarily involves the change in the effective diameter of the nanochannel before and after recognition. During the sensing process, when voltage is applied on both sides of CBA@AAO, ions are driven to

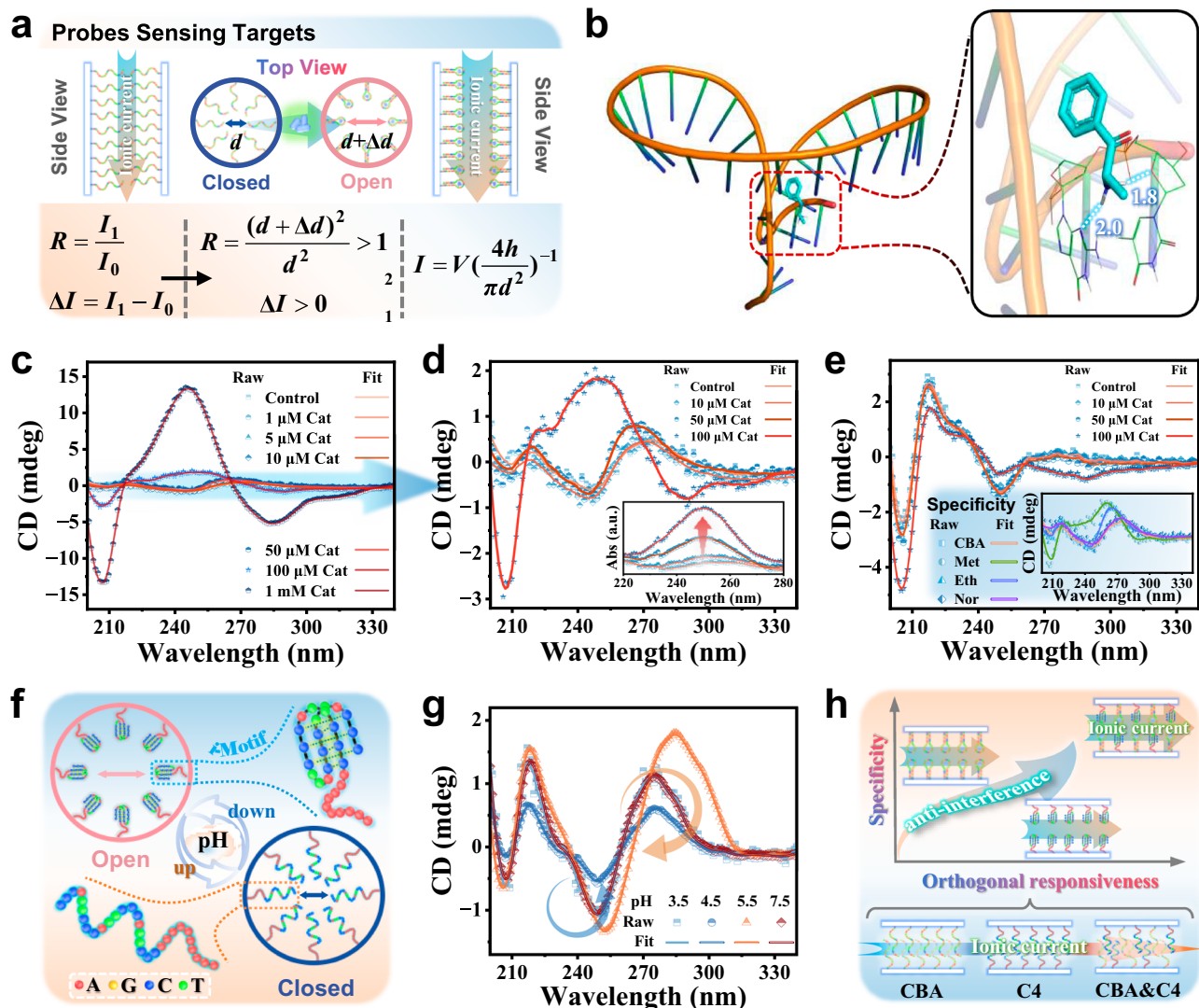

**Fig. 4 | Sensing mechanism of $N_CB_S$ modified by DNA probes. a** Cross-sectional and top views of the $N_CB_S$ based on nanochannels. Before target recognition, the CBA is stretched to block the nanochannel (closed state); after target recognition, the probe binds to the target and folds into a hairpin structure, opening the nanochannel (open state). As a result, the effective diameter of the nanochannel increases from $d$ to $d + \Delta d$. **b** Molecular docking structure of the DNA probe CBA and Cat interacting via groove binding. The white dashed lines indicate hydrogen bonds. **c** Circular dichroism (CD) spectra of CBA recorded at room temperature before and after binding with various concentrations of Cat, scanned from 340 to 200 nm. **d** Magnified CD spectra of CBA at lower Cat concentrations (10, 50, and 100 μM), with the inset displaying the corresponding UV–visible absorption spectra. **e** CD spectra of PolyA DNA before and after binding with different concentrations of Cat. The inset shows the CD spectra of CBA before and after interaction with structurally similar drugs (Met, Eth, and Nor). **f** Schematic diagram of the structural transition of C4@AAO with changing pH. **g** CD spectra of C4 DNA in Tris-HCl solution at pH values of 3.5, 4.5, 5.5, and 7.5, indicating the transition of C4 DNA from a long-chain relaxed stable state to an i-Motif structure stable state. **h** Schematic diagram of co-calibration of Cat and pH with dual-DNA probes $N_CB_S$.

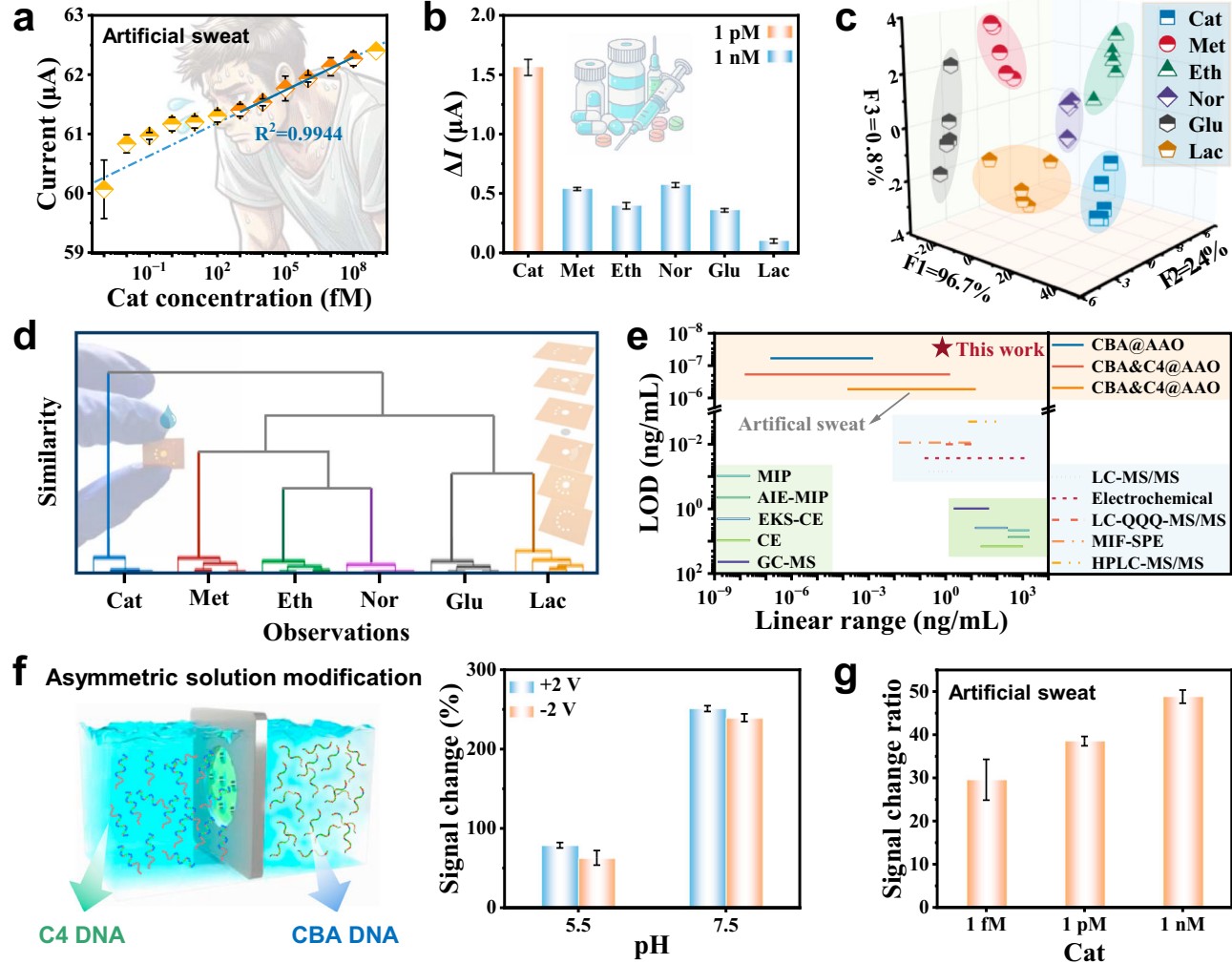

**Fig. 5 | Sensing of CBA&C4@AAO in artificial sweat based on co-calibration strategy. a** The magnitude of current response of CBA&C4@AAO to different concentrations of Cat in artificial sweat at +2 V. **b** Specificity detection of the $N_CB_S$ in artificial sweat. The concentration of Cat was 1 pM and the concentration of other disruptors was 1 nM. (Cat cathinone, Met methcathinone, Eth ethcathinone, Nor norketamine, Glu glucose, Lac lactate). **c** LDA and **d** HCA score plots were used to distinguish between four drugs and two disruptors in artificial sweat. **e** Comparison of detection limits in recent studies on Cat detection. The results show that this work is 4–5 orders of magnitude more sensitive to the detected Cat than the existing studies. Moreover, the $N_CB_S$ has a wider linear response range to the Cat,

and the detection limit in artificial sweat is as low as $5.34 \times 10^{-7}$ ng/mL. **f** Current signal changes of the CBA&C4@AAO after recognizing 1 nM Cat under pH conditions of 5.5 and 7.5, using the asymmetrically modified solution method. The left image is a schematic of asymmetric solution modification of two DNA probes. **g** An asymmetric solution-based CBA&C4@AAO identifies the ratio of current signal changes after different concentrations of Cat in an artificial sweat system. Data in the dot plot (**a**) and the bar plots (**b**, **f**, **g**) are presented as mean ± standard deviation values derived from results of three independent measurements ($N = 3$). The error bars represent standard deviation values.

pass through the nanochannel, resulting in an open-circuit current signal. The current $I$ of the nanochannel can be mathematically represented by formula (1)[59], as the system can be reduced to a model of ionic conductance for a cylindrical channel:

$$I = V[(\eta_+\mu_+ + \eta_-\mu_-)e]\left(\frac{4h}{\pi d^2} + \frac{1}{d}\right)^{-1} + \frac{V\mu \otimes \pi d\sigma}{h} \quad (1)$$

According to previous studies[52], when the thickness $h$ of the nanochannel is significantly greater than its diameter $d$, formula (1) can be simplified to the following equation:

$$I = V\left(\frac{4h}{\pi d^2}\right)^{-1} \quad (2)$$

Where $V$ is the applied voltage; $n_+$ and $n_-$ represent the number densities of positive and negative ions, respectively; $\mu_+$ and $\mu_-$ represent the electrophoretic mobilities of positive and negative ions, respectively; $e$ is the elementary charge; $h$ and $d$ are the thickness and diameter of the nanochannel. The last term in formula (1) explains the transport of ions through a highly charged inner surface, where $\mu$ represents the solution mobility of counterions adsorbed on the charged surface, and $\sigma$ is the surface charge density, with its sign opposite to that of the counterions. Equation (2) indicates that the diameter $d$ plays a crucial role in determining the current through the nanochannel. In this work, $d$ represents the effective diameter. Figure 4a discusses the relationship between $I$ and the $d$ before and after the CBA@AAO sensor recognizes the target. Before target recognition, the probe is stretched and in a dispersed state, leading to obstruction of the nanochannel current. Upon binding with Cat, the CBA probe adopts a hairpin structure, resulting in an increase in nanochannel

current. The current ratio ($R$) and the current change ($\Delta I$) for target recognition by the sensor can be expressed as follows:

$$R = \frac{I_1}{I_0} \tag{3}$$

$$\Delta I = I_1 - I_0 \tag{4}$$

Where $I_1$ and $I_0$ represent the current values measured before and after the binding of CBA@AAO with Cat, respectively. By combining Eqs. (2), (3), and (4), the current ratio and current change for the probe-$N_CB_S$ upon target recognition can be calculated as follows:

$$R = \frac{(d + \Delta d)^2}{d^2} > 1 \tag{5}$$

$$\Delta I > 0 \tag{6}$$

The significant increase in current observed after Cat binds to the CBA@AAO can be attributed to the increase in the $d$ of the nanochannel. To further demonstrate the specificity of the $N_CB_S$, we conducted CD tests using PolyA in response to different concentrations of Cat. The results indicated that PolyA did not show significant changes in CD signal due to the presence of Cat (Fig. 4e). Additionally, we examined the CD properties of Met, Eth, and Nor, which have similar structures to Cat, when interacting with CBA. The experimental results showed slight changes in the CD signals for Met and Eth.

Under higher pH conditions, C4 DNA adopts an extended linear conformation, partially blocking the nanochannels. As the pH decreases, the aptamer senses and binds to protons, causing C4 DNA to fold into an i-Motif structure, thereby increasing the effective pore size of the nanochannels (Fig. 4f). This structural transition was confirmed by molecular dynamics simulations and CD spectroscopy (Supplementary Figs. 31, 32). As shown in Fig. 4g, the CD characteristics remain the same at pH 3.5 and 7.5, indicating similar probe conformations. This is the most stable conformation of DNA under physiological conditions, with B-form DNA showing a negative peak around 250 nm and a positive peak around 275 nm[60,61]. When the pH drops from 5.5 to 4.5, the CD results show the negative peak shifting from 253 nm to 250 nm and the positive peak from 285 nm to 275 nm, indicating a structural transition from B-form to Z-form and back to B-form, leading to a final conformational change[62–64]. Figure 4h illustrates the response mechanism of the dual-aptamer calibrated $N_CB_S$ for pH-interference-resistant Cat detection (Supplementary Fig. 33). C4 DNA binds to hydrion, reducing signal interference from mass transport, while CBA provides target specificity. By minimizing pH-induced ionic current interference, the CBA&C4@AAO significantly enhances sensitivity to Cat under different pH conditions (Supplementary Figs. 34, 35).

### Application of co-calibration strategy in complex system

Interference-free detection of target substances in complex environments is a key factor in evaluating sensor performance. For drug detection through sweat, this non-invasive method offers significant potential due to its lack of privacy concerns. However, since human sweat typically has a pH range of 3–8, pH-related signal interference remains a challenge for sensors in practical applications. To address this, we used the CBA&C4@AAO to detect various concentrations of Cat in artificial sweat. As shown in Fig. 5a, the $N_CB_S$ exhibited a strong linear response between 1 pM–100 nM, with $R^2 = 0.9944$. We selected Met, Eth, and Nor, which are structurally like the Cat, as reference, while glucose (Glu) and lactate (Lac), which are common in sweat, were selected as disruptors. The results showed that the current change after incubation with Cat was 1.56 µA, while the signal change for other

substances was no more than 36.32% of the target signal (Fig. 5b). This indicates that the $N_CB_S$ produces distinct electrochemical responses for different analytes, forming the basis for discrimination. To further evaluate its discriminative capability, linear discriminant analysis (LDA) and hierarchical cluster analysis (HCA) were performed. The LDA results successfully differentiated the six substances in the complex mixture with an accuracy of 97% (Fig. 5c and Supplementary Table 13). The HCA dendrogram visually conveyed the relationships between analytes, effectively illustrating the chemical similarities (Fig. 5d).

Additionally, we evaluated the spiked recovery performance of CBA&C4@AAO in various water samples, achieving recovery rates ranging from 97.3 ± 0.9% to 102.9 ± 0.9% (Supplementary Fig. 36). The results demonstrate the excellent performance of the $N_CB_S$ in complex water matrices. Moreover, to compare the sensitivity of the $N_CB_S$ prepared through mixed modification, we further employed asymmetric modification by immobilizing C4 DNA and CBA on opposite sides of the nanochannels (Fig. 5f). At +2 V, the $N_CB_S$ current signal increased by 78.8 ± 3.4% at pH 5.5 and by 251.0 ± 3.7% at pH 7.5 after detecting 1 nM Cat, indicating that asymmetric modification enhances sensitivity under acidic conditions. We also evaluated the $N_CB_S$ response to different concentrations of Cat in artificial sweat. As shown in Fig. 5g, the signal change rates increased with Cat concentration, reaching 29.6 ± 4.7%, 38.5 ± 1.1%, and 48.8 ± 1.5% at 1 fM, 1 pM, and 1 nM, respectively. These results demonstrate that the CBA&C4@AAO prepared via this functionalization strategy maintains excellent recognition ability for Cat in complex environments. The dual aptamer-functionalized $N_CB_S$ utilizes a co-calibration strategy for Cat recognition and pH interference resistance. This sensing strategy not only achieved a wider target concentration response range but also significantly improved the detection limit by 4–5 orders of magnitude compared to recent drug detection methods (Fig. 5e and Supplementary Table 14). Thus, it holds great promise for drug detection in sweat, drug control, and health monitoring.

## Discussion

In summary, we propose a co-calibration strategy with biomimetic solid-state $N_CB_S$ with dual-DNA probes. The CBA as AG, calibrates ultrasensitive and selective Cat recognition. The C4 DNA as SF, calibrates selective ions transport for pH response. MD results show that hydrogen bonding and van der Waals forces drive the groove binding between Cat and the CBA. Compared with other drugs with similar chemical structures, this $N_CB_S$ exhibits good selectivity and sensitivity towards Cat. An excellent linear relationship was observed between the target concentration and the output ionic current ratio in the range of 1 fM-10 pM, with a LOD as low as 0.40 fM. Furthermore, the $N_CB_S$ shows good cross-reactivity in the analysis of multiple drugs, and CBA@AAO can distinguish fourteen types of drugs with 100% accuracy. It indicates that the $N_CB_S$ has potential applications in the discrimination and analysis of multiple analytes in complex environments. To address the interference caused by different pH levels in sweat sensing, we introduced a pH-responsive DNA probe and designed dual-probes co-calibration $N_CB_S$ (CBA&C4@AAO). A wider range of linear responses at different pH values is achieved. The detection results in artificial sweat show that CBA&C4@AAO has good anti-interference recognition ability to Cat and has excellent linear response ability at 1 pM-100 nM, and the LOD is as low as 3.58 fM. Overall, the dual-probe functionalized $N_CB_S$ offers co-calibration for multi-target recognition, confirming better sensitivity and anti-interference for target recognition compared to traditional single-probe sensors. Co-calibrated probes $N_CB_S$ provides an approach for designing ultra-sensitive and selective drug detection biosensing devices for drug monitoring, sweat sensing, and healthcare.

However, moving nanopore-based detection methods toward practical applications still involves overcoming several key challenges. These include enhancing detection specificity, improving membrane

stability and reproducibility, increasing sensitivity and throughput, and achieving device portability and user-friendliness. Additionally, effective data analysis and interpretation remain critical, given the complexity of nanopore signals. To address these issues, strategies include precise surface functionalization and the use of target-specific recognition elements to improve selectivity; optimizing membrane materials and fabrication processes to ensure stable and reproducible performance; and employing smaller pore diameters, parallel nanopore arrays, and optimized electrochemical conditions to boost sensitivity and throughput. Furthermore, miniaturization, integration, and automation can greatly facilitate practical usage, while the incorporation of machine learning and automated data processing tools can streamline data interpretation and enhance accuracy. The combined implementation of these strategies will help propel nanopore-based detection from the laboratory toward real-world applications, offering more efficient and reliable solutions in fields such as bioanalysis, clinical diagnostics, and environmental monitoring.

## Methods

### Materials and reagents

Anodic aluminum oxide (AAO) nanochannel membranes were procured from Hefei PuYuan Nano Technology Co., Ltd (Hefei, China). The synthesis and purification of the DNA oligonucleotides, including the cathinone-binding probe (CBA, 5'-NH$_2$-(CH$_2$)$_6$-ACT GAG AAG TGT GAT TCA GTA TGT TTT CCG AAG T-3') and the pH-responsive probe (C4 DNA, 5'-NH$_2$-(CH$_2$)$_6$-AAA AAA AAA ACC CTT ACC CTT ACC CTT ACC C-3'), were carried out by Sangon Biotech Co., Ltd. (Shanghai, China). Detailed sequences related to other experiments are listed in Supplementary Table 1. NaCl (99.5%) and MgCl$_2$ (hexahydrate, 98%) were purchased from Shanghai Macklin Biochemical Co., Ltd. Additionally, 25% glutaraldehyde (GA) aqueous solution, isopropanol (≥95%), and sucrose (Sul, 99%) were also obtained from Macklin. (3-aminopropyl) trimethoxysilane (APTES, >98%), Lactose (Lac, D-(+)-Lactose), and mannitol (Man, D-Mannitol) were acquired from Aladdin (Shanghai, China). Hydrochloric acid (HCl, 36.0% ~38.0%) was supplied by Guangzhou Chemical Reagent Factory.

The stock solutions of drugs (1.0 mg·mL$^{-1}$), including cathinone (Cat), methcathinone (Met), ethcathinone (Eth), norketamine (Nor), ketamine (Ket), amphetamine (Amp), (±)-3,4-methylenedioxymethamphetamine (MDMA), phenacetin (Phe), cocaine (Coc), heroin (Her), caffeine (Caf), procaine (Pro), paracetamol (Par), and (+)-pseudoephedrine ((+)-PSE) were legally provided by criminal evidence samples of the institute of evidence laboratory at China University of Political Science and Law (CUPL) and Guangdong Province Key Laboratory of Psychoactive Substances Monitoring and Safety. The analytical samples were prepared by diluting the stock solutions in either 0.01 M Tris-HCl buffer or artificial sweat. Artificial sweat (pH 7.3) was purchased from YuanYe Biotechnology Co., Ltd (Shanghai, China). Unless otherwise specified, all chemical reagents were of analytical grade and used without further purification. All solutions were prepared using ultrapure water produced by a Millipore Milli-Q system (18.2 MΩ·cm).

### Fabrication of functionalized N$_C$B$_S$

The AAO membranes with a pore size of 40–70 nm were sonicated in ultrapure water for 3 min. After drying, the membranes were treated with anhydrous ethanol for 2 h to ensure that the surfaces and internal channels of the AAO membranes were thoroughly cleaned. At RT, the AAO membrane surfaces were aldehyde-functionalized (GA@AAO) using a two-step chemical method. Subsequently, 1 µM CBA was grafted into the nanochannels via Schiff base reaction, resulting in CBA@AAO. Prior to -CHO modification, the AAO nanochannel membranes were treated with a (3-aminopropyl) trimethoxysilane (APTES, 15%) isopropanol solution for 12 h to aminate the membrane surface (-NH$_2$). Then, the membranes underwent a Schiff base reaction with

glutaraldehyde (GA, 15%) aqueous solution for 12 h to successfully modify the surface of the nanochannel membranes with -CHO groups. A similar method was employed to prepare the pH-responsive N$_C$B$_S$ (C4@AAO), as well as the dual-probes nanochannels with co-calibration to Cat and pH (CBA&C4@AAO). For the preparation of CBA&C4@AAO using the asymmetric solution method, GA@AAO was placed in a custom mold, and equal amounts of 1 µM CBA and C4 DNA were added to the two separate electrolytic cells on either side of the membrane.

### Current-voltage measurement and real-time current recording

The samples were installed in a dual-chamber electrochemical cell and measured using a Keithley 6487 picoammeter (Keithley Instruments, USA). Ionic current measurements were conducted in a 0.01 M Tris-HCl (pH 7.4) electrolyte solution using a custom-made Ag/AgCl electrode. The effective area for ion conduction was ~5 mm$^2$. The voltage was scanned between −2 V ~ + 2 V with a step voltage of 0.1 V to plot the current-voltage (I-V) curves. A constant voltage of +2 V was applied, and real-time current (I-T) recordings were taken over a period of 60 s. All tests were conducted at RT, with each test performed at least three times in parallel to obtain average values.

### Apparatus and characterization

SEM was performed using the Zeiss GeminiSEM 360, operating at an accelerating voltage of 5.00 kV (at magnifications of 50,000× and 100,000×), to characterize the surface morphology of the nanochannels. CA measurements were conducted using the SDC-350 from Shengding Precision Instrument Co., Ltd. (Dongguan, China). For each measurement, 4 µL of ultrapure water (18.2 MΩ·cm) was dropped onto the membranes surface at RT for investigation. EDS mapping was performed using a Hitachi/SU1000 high-resolution SEM to qualitatively analyze the distribution of carbon (C), nitrogen (N), oxygen (O), and silicon (Si) elements on the membranes surface before and after nanochannel functionalization. Subsequently, XPS was employed on a Thermo Scientific™ K-Alpha™⁺ spectrometer to quantitatively test the C$_{1s}$, N$_{1s}$, and O$_{1s}$ on the membranes surface before and after nanochannel functionalization. The testing software used was Thermo Avantage v5,9921.

CD spectra were acquired using a Chirascan Plus spectropolarimeter (Applied Photophysics, UK) over the wavelength range of 200–340 nm, employing 10 mm path-length quartz cuvettes. The test samples comprised 1 µM CBA I with varying concentrations of Cat, 1 µM PolyA I with different concentrations of Cat, 1 µM CBA I and PolyA I with varying concentrations (10 and 100 µM) of Cat, Met, and Eth, and C4 DNA I incubated for 2 h in 0.01 M Tris-HCl solutions at different pH levels (3.5, 4.5, 5.5 and 7.5). Additionally, a blank group was prepared by mixing 1 µM CBA I with 100 µM Met, Eth, and Nor respectively, followed by the addition of varying concentrations of Cat and incubation for at least 2 h to investigate the response to Cat in complex systems. Background scans of 0.01 M Tris-HCl (pH 7.4) buffer were initially conducted, and the blank background was subtracted from the sample spectra after each collection. All tests were conducted at RT.

### Numerical simulation

Molecular docking: Referring to our previous work[65], computational-assisted compound docking was conducted using the Maestro interface within the Schrödinger 2021-3 software package. The structure of Cat bound to the appropriate DNA (CBA) was constructed using RNAComposer (http://rnacomposer.cs.put.poznan.pl/), while the ligand (Cat) was designed using ChemDraw. Further refinement was performed using the PyMOL Molecular Graphics System 3.0 (The PyMOL Molecular Graphics System, Version 3.0 Schrödinger, LLC.) for docking computations. Prior to docking, the LipPrep module was employed to compute the charge state and optimal structure for the binding free energy of Cat under physiological conditions. The

receptor grid generation module in Maestro was utilized to generate grid files defining the docking interface, with the center of the CBA chain centroid chosen as the center of the active pocket, and a grid file with dimensions of $40 \times 40 \times 40$ Å$^3$ was generated. Finally, the ligand was docked to the receptor using Glide in SP mode.

Molecular dynamics simulation: The molecular dynamics simulations were performed using Amber24 software, with the OL21 and Constph force fields selected, and the TIP3P water model was employed. The complex was placed in a cubic water box. The cutoff distance for both electrostatic and van der Waals interactions was set to 1.0 nm, and a time step of 2 fs was used. The long-range correction for electrostatic interactions was applied using the PME method. The system was maintained at 300 K with a pressure of 1 bar and a pH of 3 and 8. One composite system was constructed in total. After constructing the system, energy minimization was first performed, followed by 200 ps of NVE equilibrium dynamics and 100 ps of NPT equilibrium dynamics. The system temperature was coupled to a heat bath using the V-rescale method, and pressure was controlled using the Parrinello-Rahman method. Finally, a 50 ns molecular dynamics sampling was carried out. Relevant indicators, such as RMSD, RMSF, radius of gyration, and hydrogen bond count, were calculated using Amber's built-in modules, Cpptraj, and Python scripts.

### Data analysis
SEM images and CA measurements were processed using Adobe Photoshop 22.1.1. For nanochannel pore size and distribution statistics of the AAO membranes, Nano Measurer 1.2 software from the Surface Chemistry and Catalysis Laboratory at Fudan University was employed. Pore density statistics and CA measurements utilized Image J 1.51j8 (Wayne Rasband, National Institutes of Health, USA). Data plotting was performed with Origin 2024 (OriginLab Co., Northampton, USA). Thermo Scientific™ Avantage v6.6.0 was utilized for charge correction in XPS data, setting the standard value for external carbon contamination at 284.8 eV for the $C_{1s}$ peak. PCA was conducted using SYSTAT 13 (Systat Software Inc., Chicago, IL, USA), and HCA was executed with Minitab 19 (Minitab Inc., State College, PA). Quantitative identification of Cat involved a limit of detection (LOD) calculation:

$$LOD = \frac{3.3 \times \delta}{S}$$

where $\delta$ represents the standard deviation of response values from multiple measurements of the blank group, and $S$ is the slope of the standard curve. All data were obtained from at least three independent parallel experiments, and error bars indicate standard deviation (SD) values.

### Reporting summary
Further information on research design is available in the Nature Portfolio Reporting Summary linked to this article.

## Data availability
All relevant data that support the findings are available within this article and Supplementary Information. A reporting summary for this Article is available as a Supplementary Information file. Please follow the link for data download: https://figshare.com/s/b65665f8ecaf8b4c4102 Source data are available for Figs. 2, 3, 4c–e, 4g, 5a–d and 5f,g and Supplementary Fig. 1–3, 5, 6, 8, 10–20, 22–32 and 34–36 in the associated source data file.

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

## Acknowledgements

This work is supported by the National Natural Science Foundation of China (22474049 to F.L., 52003103 to F.L., 22090050 to Y.H., and 22176180 to Y.H.), the National Key R&D Program of China (2016YFC1100502 to F.L., and 2021YFA1200400 to Y.H.), the Key research and development project of Guangdong Province (Grant No.2020B010190003 to F.L.), and Natural Science Foundation of Zhejiang Province (Grant No. LD21B05000 to F.L.). Thanks eceshi (www.eceshi.com) for the XPS test.

## Author contributions

Y.C. and F.L. conceived and designed the experiments. Y.C., X.L., X.Y., W.Y., Y.S. and Z.H. performed the measurements and experiments. Y.C., X.Y., Y.W., Y.H., F.X. and F.L. contributed to the data analysis. Y.C., Y.H., F.X. and F.L. wrote the manuscript. All authors reviewed and commented on the manuscript.

## Competing interests

The authors declare no competing interests.
