## [Transparent Peer Review file · Nature Communications]

Sub-femtomolar drug monitoring via co-calibration mechanism with nanoconfined DNA probes

Corresponding Author: Professor Fengyu Li

Version 0:

Reviewer comments:

Reviewer #1

(Remarks to the Author)

In the manuscript entitled "Sub-Femtomolar Drug Monitoring via Co-Calibration Mechanism with Nanoconfined DNA Probes", Yonghuan Chen and coworkers propose a co-calibration mechanism for DNA probes in a nanoconfined channel, enabling sub-femtomolar drug monitoring in sweat range pH3.0~8.0. This strategy synergistically combines surface wettability regulation and hydrion capture through the modification of dual-DNA probes to enhance sensor sensitivity. Moreover, the detection limit reaches as low as 3.58 fM in artificial sweat, with a broad linear response range from 103 to 108 fM. This co-calibration nanoconfined channel offers a platform for enhanced target recognition sensitivity, showing considerable potential in sweat sensing and drug monitoring. Overall, the topic of this manuscript is fascinating, and the manuscript structure presented here is comprehensive. For those reasons, it is recommended to publish this manuscript after a minor revision.

1. On page 2, the authors stated "However, changes in pH directly affect the hydration state of ions and their transport behavior within the nanopore." the authors are encouraged to summarize the research about modulating the pH to modify the interaction network within the confined space (Angew. Chem. Int. Ed., 2021, 60, 24582-24587) and the finding that highly charged nanopores could slow or reverse protein transport based on pH (Nano Lett. 2010, 10, 6, 2162-2167).

2. On page 6, the authors stated "To achieve co-calibration for ultra-sensitive Cat detection and pH interference resistance, we developed the CBA&C4@AAO by co-grafting CBA and C4 DNA onto NcBs." Both modified molecules are DNA. The authors should explain how to control the modification ratio of the two molecules in the experiment.

3. In Figure 1, the author states, 'Figure 1. The Co-Calibration of Dual-DNA Probes in Nanoconfined Channel (CBA&C4@AAO) for Drug Monitoring and Anti-pH Interference to Improve the Wide-Range Linear Response and Ultra-Sensitivity.' It seems likely that the author intended to say 'Drug Monitoring' rather than 'Drug Monitoring'.

4. In Figure 2, the results in C and D differ from A and B. The authors should explain this discrepancy.

5. Figure 3F lacks error bars. The authors are advised to include them.

6. The author should revise Figure 3, as some pH data points are quite difficult to distinguish. A similar issue also appears in Figures 4C-E.

7. Why are there no values on the y-axis for the circular dichroism spectra in Figure 3?

8. For all the Figures in the manuscript, the authors are encouraged to minimize the use of background color to better highlight the experimental data.

9. At present, nanopore-based detection is still in the early stages of development. The authors are encouraged to summarize the challenges and corresponding strategies for advancing the practical application of this method in the discussion section.

10. How is the modification of DNA probes controlled in nanoconfined biosensors? Is the modification of DNA probes excessive? Are both the outer and inner surfaces modified? How is the reproducibility?

Reviewer #2

(Remarks to the Author)

To address the challenge of pH interference in sweat drug detection, the authors propose a nanoconfined biosensor based on a multi-DNA probe co-calibration strategy. This innovative approach combines the high responsiveness of single DNA probes with enhanced signal amplification within a nanoconfined environment, enabling ultra-sensitive drug monitoring across a sweat pH range of 3 to 8. This strategy offers a significantly broader linear response range and enhances sensitivity

by 4–5 orders of magnitude. In general, this paper succeeded to show innovation and convincing data analysis. The topic is interesting, and the data is exceptionally plentiful. This paper is well prepared. However, minor revisions are recommended before it can be considered for publication in this journal.

1. The design concept of this study is inspired by the distribution of different functional components in biological nanopores. However, this idea is not clearly illustrated in Figure 1. It is suggested to further explain the core concept of the design and provide additional details on how the distribution of different functional components enables efficient multi-DNA probe calibration and enhanced signal recognition. Consider adding schematic diagrams or flowcharts in the figure to more clearly convey this innovative design concept.

2. The study emphasizes the introduction of multi-DNA probes, which, compared to a single DNA probe, enhances detection sensitivity primarily through changes in the wettability of the nanochannels surface. However, the manuscript lacks detailed experimental data to support this aspect of the research. It is recommended to include relevant experimental data that demonstrate the specific characterization of wettability changes on the nanochannels surface after modification with multi-DNA probes. This will enhance the reliability and persuasiveness of this section of the study.

3. This work selects two different DNA probes. One is a target-specific recognition aptamer, whose binding interaction with the target has been demonstrated through molecular docking experiments. The other is a pH-specific DNA probe, for which no related theoretical simulation has been conducted, and the data only provides CD spectroscopy. Please provide a detailed explanation of the scientific basis and reliability of the design of these DNA probes.

4. Currently, the XPS characterization data only shows the functionalization of a single DNA probe on the nanopore, and lacks data for the mixed modification of both DNA probes. It is recommended to supplement the XPS data for the co-modification of the two DNA probes and compare it with the results of the single DNA probe. This would not only validate the increase in the surface modification density of the nanopore functionalized with dual aptamers, but also further confirm the effectiveness and advantages of the dual-DNA probe co-calibration strategy.

5. The color scheme of the images in the manuscript is generally clear, but some images (such as Figure 4A) require further enhancement of color contrast and clarity to ensure that the details in the image are more prominent and easier to distinguish. It is recommended to adjust the colors accordingly. In addition, the format of the references should comply with requirements of the journal. For example, "ACS Nano 18 (8), 6570-6578 (2024)." and "J. Am. Chem. Soc 146, 20, 14058-14066 (2024)." in the references. It is suggested to recheck and ensure that all references are consistent in format. In addition, the details of the manuscript presentation need to be carefully checked, for example, the reference to "supporting information" in the manuscript should be changed to "supplementary information".

Version 1:

Reviewer comments:

Reviewer #1

(Remarks to the Author)

The authors have done a very thorough job responding to the comments and have updated their manuscript accordingly. I recommend publication without further revision, and this will be a nice addition to Nature Communications

Reviewer #2

(Remarks to the Author)

The authors have fully addressed my previous comments.

Response to Comments of the Reviewers

We appreciate the constructive comments and suggestions provided by the reviewers. In response, we have revised the manuscript accordingly as detailed in the responses below. All corresponding changes have been highlighted in blue in the main text and the supplementary information for ease of reference.

◆ Reviewer #1 (Remarks to the Author):

In the manuscript entitled “Sub-Femtomolar Drug Monitoring via Co-Calibration Mechanism with Nanoconfined DNA Probes”, Yonghuan Chen and coworkers propose a co-calibration mechanism for DNA probes in a nanoconfined channel, enabling sub-femtomolar drug monitoring in sweat range pH3.0~8.0. This strategy synergistically combines surface wettability regulation and hydrion capture through the modification of dual-DNA probes to enhance sensor sensitivity. Moreover, the detection limit reaches as low as 3.58 fM in artificial sweat, with a broad linear response range from 10^3 to 10^8 fM. This co-calibration nanoconfined channel offers a platform for enhanced target recognition sensitivity, showing considerable potential in sweat sensing and drug monitoring. Overall, the topic of this manuscript is fascinating, and the manuscript structure presented here is comprehensive. For those reasons, it is recommended to publish this manuscript after a minor revision.

Response: We sincerely appreciate the valuable comments provided by the reviewer and the acknowledgment of our diligent efforts. We have carefully considered and addressed all the concerns of the comments. Corresponding revisions have been incorporated into both our manuscript and supplementary information, with all changes highlighted in blue. Thank you for your constructive feedback.

Comments 1: On page 2, the authors stated “However, changes in pH directly affect the hydration state of ions and their transport behavior within the nanopore.” the authors are encouraged to summarize the research about modulating the pH to modify the interaction network within the confined space (*Angew. Chem. Int. Ed.*, **2021**, 60, 24582-24587) and the finding that highly charged nanopores could slow or reverse protein transport based on pH (*Nano Lett.* **2010**, 10, 6, 2162-2167).

Response: Thank you for your careful review of our work and insightful suggestion. In response, we have incorporated a summary of relevant research that demonstrates how pH modulation influences the interaction network within confined nanochannels [*Angew. Chem. Int. Ed.* **60**, 24582-24587 (2021)]. Variations in pH can directly impact the hydration states of ions and their transport behavior within the nanopore and even modify the interaction network within the confined space. Consequently, shifts in pH alter the surface charge density of the nanopore walls and the distribution of electric potential, thereby affecting the local arrangement of water molecules and associated hydration state. Under these conditions, highly charged nanopores may slow or even reverse protein transport based on pH, further underscoring the critical role of pH in governing selective transport phenomena. [*Nano Lett.* **10**, 6, 2162-2167 (2010)]. Such alterations have profound

implications for target recognition and sensitivity, especially when detecting complex biological samples like sweat, where the pH fluctuates between 3.0~8.0. These points have been included in the revised manuscript to strengthen our discussion on the impact of pH in ion hydration and transport behavior. We have highlighted the modifications made in blue within our revised version. In addition, we have added references 43 and 46. Please refer to page 2 and page 16 of our revised manuscript for relevant revisions. Thank you for your review.

Comments 2: On page 6, the authors stated “To achieve co-calibration for ultra-sensitive Cat detection and pH interference resistance, we developed the CBA&C4@AAO by co-grafting CBA and C4 DNA onto NcBs.” Both modified molecules are DNA. The authors should explain how to control the modification ratio of the two molecules in the experiment.

Response: Thank you for your professional and technical insight. We controlled the modification ratio of the two molecules to 1:1 in this work, and the relevant details have been noted in the revised manuscript. In addition, we have conducted additional experiments to explore the target responsiveness of functionalized nanochannels of different proportions of DNA molecules (**Figure R1**). The experimental results show that the ionic current of NcBs modified with different proportions of DNA molecules does not show any directional trend (**Figure R1A, B**), which may be influenced by many aspects. As far as we know, the directional current generated by ion transmembrane transport is mainly affected by the synergistic effects of surface wetting, surface charge and effective size, and this process is very complicated [*Angew. Chem. Int. Ed.* **61**, e202207369 (2022)]. This was also confirmed by contact Angle measurements, and the changes in surface wettability did not follow a corresponding regularity (**Figure R11**). However, the overall results showed that the hydrophilicity of double-DNA molecule comodification was slightly lower than that of single-DNA molecule modification, and when the ratio of two molecules was 1:1, the hydrophilicity was the lowest in the test group. Thus, this may enhance the response performance of the NcBs to the target at this ratio (**Figure R1C**). We have made a complete supplement to the revised manuscript and the supplementary information, please see page 6 of the revised manuscript and pages 23,43,44 of the supplementary information. We appreciate the reviewer’s valuable feedback, and we will continue to strengthen the research in this direction in the follow-up work.

Figure R1 (Supplementary Figs. 30, 31). Ionic current signal changes following nanochannel functionalization with varying CBA:C4 DNA ratios. (A) The ionic current of DNA molecules with different proportions at +2 V before and after functionalization. **(B)**

Current changes before and after functionalization. (C) Current changes of NcBs modified with different proportions of DNA molecules before and after response to 1 pM target.

Comments 3: In Figure 1, the author states, “Figure 1. The Co-Calibration of Dual-DNA Probes in Nanoconfined Channel (CBA&C4@AAO) for Drug Monitoring and Anti-pH Interference to Improve the Wide-Range Linear Response and Ultra-Sensitivity.” It seems likely that the author intended to say “Drug Monitoring” rather than “Drug Monitoring”.

Response: We appreciate the reviewer for pointing out this typographical error. We have corrected “Drug Monitoring” to “Drug Monitoring” in the caption of **Figure 1**. Please refer to the drawing note to **Figure 1** on page 3 of the revised manuscript

Comments 4: In Figure 2, the results in C and D differ from A and B. The authors should explain this discrepancy.

Response: Thanks for the reviewer’s comment. Herein, we examined two detection strategies—incubation-based detection (**Figures 2A, B**) and direct detection (**Figures 2C, D**)—and observed that their transmembrane ionic current responses exhibited opposite trends (**Figure R2**). In the incubation-based method, upon binding 1 pM cathinone to CBA@AAO, the transmembrane ionic current increased. This phenomenon can be attributed to a conformational shift in the CBA probe from an extended chain to a hairpin structure upon target binding, effectively enlarging the functionalized nanochannel aperture and thus enhancing ionic flux (**Figure R3A**). The *I-T* characteristic tests further supported this finding, revealing a more pronounced current signal following the conformational change (**Figure R3A-①**). Moreover, both the rapid hypersensitive *I-V* responses and the stable *I-T* data confirm that incubation-based cathinone detection is achieved by modulating the effective size of the nanochannel, thereby influencing ionic transport through the membrane (**Figure R3A-②**).

Figure R2. Changes in the electrical signals of CBA@AAO following two detection approaches—2-hour incubation (I) and direct detection (II)—in the presence of 1 pM cathinone. The inset shows the current changes before and after cathinone response.

Experimental results from the direct detection of cathinone transmembrane transport revealed a decreasing *I-T* signal as the cathinone concentration increased. As shown in **Figure R3B (Figure 2D)**, at high cathinone levels, during the process of CBA as a specific binding probe transporting cathinone, it gradually blocks the nanochannel and shields the

negatively charged surface of the nanochannel, resulting in varying degrees of pore blockage [J. Am. Chem. Soc. **133**, 17307-17314 (2011)]. In our revised manuscript, we have provided further clarification and improvements, which are highlighted in blue. We appreciate your thoughtful review.

Figure R3. (A) Schematic illustration of the CBA@AAO structure transition into a hairpin configuration upon cathinone binding, resulting in an enlarged effective nanochannel aperture. ① I - T characteristic of CBA@AAO binding 1 pM cathinone within 60 s before and after interaction; ② Comparison of the rapid-response (I - V) and stable-response (I - T) signals of CBA@AAO at +2 V in the presence of 1 pM cathinone. (B) I - T characteristics of CBA@AAO during direct detection of varying cathinone concentrations, accompanied by a schematic depicting the proposed transport mechanisms.

Comments 5: Figure 3F lacks error bars. The authors are advised to include them.

Response: We thank the reviewer for professional assessment and detailed feedback. Initially, we considered adding error bars to the 3D data graph in **Figure 3F** but found that this approach did not yield a clear or visually effective representation. To address this concern, we have included a 2D version of the same data in the Supplementary Information (**Supplementary Figure 34D**), where each data point is accompanied by clearly visible error bars (**Figure R4**). We take data reproducibility very seriously, and each data set presented is derived from at least three independent parallel experiments.

Figure R4 (Supplementary Fig. 34D). Ionic current variations of CBA&C4@AAO in response to varying cathinone concentrations under different pH conditions.

Comments 6: The author should revise Figure 3, as some pH data points are quite difficult to distinguish. A similar issue also appears in Figures 4C-E.

Response: Thank you for highlighting this issue. In the revised manuscript, we have improved the figure designs to enhance overall clarity (**Figures R5, R6**). Specifically, we have adjusted the color schemes so that each pH condition is easily distinguishable (**Figures 3A, D**). Herein, the C4@AAO exhibited significant variations in *I-V* characteristics in response to different pH levels (**Figures R5A**). At +2 V, the transmembrane current varied notably: $85.6 \pm 1.3 \mu\text{A}$ at pH 3 (open state) and $20.2 \pm 0.1 \mu\text{A}$ at pH 8 (closed state). The C4-NcBs also showed a strong linear response between pH 3 ~ 7 ($R^2 = 0.9881$) (**Figures R5B**), attributed to pH-induced structural changes in C4 DNA. We also investigated the ionic current changes of CBA@AAO and CBA&C4@AAO before and after target binding under different pH conditions (**Figures R5D**). The current measurements were based on *I-V* characteristics, reflecting the transmembrane ionic current changes before and after Cat recognition at +2 V. The results showed that the incorporation of C4 DNA led to a significant increase in the current variation upon target recognition, compared to CBA@AAO.

Figure R5 (Fig. 3). Sensing Properties of C4@AAO and CBA&C4@AAO at Different pH Levels. (A) *I-V* characteristics of the C4@AAO in response to different pH levels. (B) Current values of the C4@AAO at +2 V in response to different pH levels. The inset shows the linear fit of the NcBs response within the pH range of 3 ~ 7, with an R^2 value of 0.9881. (C) Cycle stability response of C4@AAO at applied voltages of ± 2 V under pH conditions of 5.5 and 7.5. (D) Based on the *I-V* characteristics, the ionic current responses

of CBA@AAO and CBA&C4@AAO to 1 nM cathinone were measured under pH conditions ranging from 3 ~ 8 at +2 V. (E) The current changes at +2 V for both CBA@AAO and CBA&C4@AAO before and after cathinone binding were further examined at pH 5.5 and 7.5. (F) The current signal responses of CBA&C4@AAO to different concentrations of cathinone under varying pH conditions, relative to the signal changes caused by pH interference. (G) The response of CBA&C4@AAO to varying concentrations of cathinone at pH 7.4. (H) At pH 5.5, the *I*-*V* current responses (+2 V) of CBA@AAO and CBA&C4@AAO to different cathinone concentrations. (I) The linear response range and a low limit of detection (LOD) of CBA@AAO and CBA&C4@AAO at two different pH levels.

Additionally, we removed the background color in **Figure 3** and further refined **Figures 4C-E** to improve their readability and clarity. Circular dichroism (CD) measurements showed that conformational changes occurred after CBA was combined with the target Cat (**Figures R6A, B**). With the increasing of Cat concentration, the change of CD signal is more obvious, and the corresponding ultraviolet absorption increases. Therefore, the proposed mechanism mainly involves changes in the effective diameter of the nanochannel before and after identification. To further demonstrate the specificity of the NcBs, we conducted CD tests using PolyA in response to different concentrations of Cat. The results indicated that PolyA did not show significant changes in CD signal due to the presence of Cat (**Figure R6C**). Moreover, we examined the CD properties of Met, Eth, and Nor, which have similar structures to Cat, when interacting with CBA. The experimental results showed slight changes in the CD signals for Met and Eth. Please see pages 7 and 9 of the revised manuscript for relevant changes. We appreciate the reviewer's constructive feedback again.

Figure R6. (A) Circular dichroism (CD) spectra of CBA recorded at room temperature before and after binding with various concentrations of cathinone, scanned from 340 to 200 nm. (B) Magnified CD spectra of CBA at lower cathinone concentrations (10, 50, and 100 μ M), with the inset displaying the corresponding UV-visible absorption spectra. (C) CD spectra of PolyA DNA before and after binding with different concentrations of cathinone. The inset shows the CD spectra of CBA before and after interaction with structurally similar drugs (Met, Eth, and Nor).

Comments 7: Why are there no values on the y-axis for the circular dichroism spectra in Figure 3?

Response: Thank you for raising this point. In order to better highlight the comparative

changes in the circular dichroism (CD) spectra before and after binding between the specific aptamer (CBA) or the nonspecific aptamer (PolyA DNA) and their respective targets (as well as target analogs), we have plotted the y-axis values of the circular dichroism in **Figures 4C-E** and **4G**. In addition, we have provided more comprehensive 3D data figures with clearly labeled CD intensity (y-values) in the Supplementary Information (**Figure R7**). These supplementary data allow for a more detailed examination of the CD signals. Please refer to page 9 of the revised manuscript for relevant changes. We express our profound appreciation to the reviewer for the dedicated review process and clear, constructive remarks that have enriched the quality and impact of our findings.

Figure R7 (Supplementary Fig. 24). (A) Changes in circular dichroism (CD) spectral signals of CBA after interaction with different concentrations of cathinone. (B) UV absorption changes of CBA after binding to different concentrations of cathinone. (C) CD signal changes of PolyA after interaction with different concentrations of cathinone. (D) CD signal changes of PolyA after interaction with 10 and 100 μM of methcathinone, ethcathinone, and norketamine. (E) CD signal changes of CBA after interaction with 10 and 100 μM of methcathinone, ethcathinone, and norketamine. (F) CD response signals of CBA to different concentrations of target cathinone in a mixed sample, where the mixed sample (blank group) consists of 1 μM CBA and 100 μM of methcathinone, ethcathinone, and norketamine.

Comments 8: For all the Figures in the manuscript, the authors are encouraged to

minimize the use of background color to better highlight the experimental data.

Response: Thank you for the reviewer's constructive suggestion. We agree that reducing background color can help emphasize the experimental data. In the revised manuscript, we have removed or minimized the use of background colors in **Figures 2, 3, and 5**, thereby enhancing the visibility and clarity of the presented data. We appreciate your feedback and believe these adjustments result in a more straightforward and reader-friendly visual presentation. Please review our revised manuscript.

Comments 9: At present, nanopore-based detection is still in the early stages of development. The authors are encouraged to summarize the challenges and corresponding strategies for advancing the practical application of this method in the discussion section.

Response: Thank you for your valuable suggestions. Advancing nanopore-based detection methods toward practical applications requires addressing several critical challenges. In the revised manuscript, we have included a dedicated section in the Discussion to outline these hurdles. These challenges include enhancing detection specificity, improving membrane stability and reproducibility, increasing sensitivity and throughput, and achieving device portability and user-friendliness. Furthermore, effective data analysis and interpretation remain pivotal due to the inherent complexity of nanopore signals.

To tackle these challenges, we propose strategies such as precise surface functionalization and the incorporation of target-specific recognition elements to enhance selectivity; optimizing membrane materials and fabrication processes to ensure consistent and stable performance; and leveraging smaller pore diameters, parallel nanopore arrays, and optimized electrochemical conditions to boost sensitivity and throughput. Additionally, miniaturization, integration, and automation of devices can significantly improve usability, while the integration of machine learning and automated data processing tools can streamline signal interpretation and improve accuracy.

By implementing these strategies, nanopore-based detection can progress from the laboratory to real-world applications, offering efficient and reliable solutions across diverse fields, including bioanalysis, clinical diagnostics, and environmental monitoring. We hope these additions provide a comprehensive roadmap for the future development of this technology. We extend our warmest thanks to the reviewer for perceptive suggestions, ensuring that our final manuscript is more robust and convincing.

Comments 10: How is the modification of DNA probes controlled in nanoconfined biosensors? Is the modification of DNA probes excessive? Are both the outer and inner surfaces modified? How is the reproducibility?

Response: Thank you for your insightful comments. The modification of DNA probes in nanoconfined biosensors is achieved through a controlled surface functionalization process (**Figure R8**). Similar functionalization methods are now very mature and have been repeatedly verified by predecessors [*Chem. Soc. Rev.* **52**, 6270-6293 (2023)]. In our experiments, we closely monitored and optimized parameters such as DNA concentration,

reaction time, and buffer conditions to ensure a reproducible and consistent grafting density (**Supplementary Figures 12, 14-16**). And we added the corresponding controlled experiments of the two DNA probes modification ratios according to the suggestion (**Supplementary Figures 30-31**). This approach helps prevent excessive DNA probe loading, as overly dense packing can cause steric hindrance, reduce accessibility to the target, or affect ion transport. Experimental results indicate that when the two DNA molecules are co-immobilized at a 1:1 ratio onto the solid-state nanochannels, no excessive accumulation of DNA probes occurs (**Figure R1**). Furthermore, solid-state nanochannels functionalized with this mixed ratio (1:1) do not exhibit a reduction in recognition signals during target detection (**Figure R9**).

Figure R8. Schematic illustration of the preparation process for the functionalized nanochannel DNA@AAO.

Figure R9. Current changes of NcBs modified with different proportions of DNA molecules before and after response to 1 pM target.

In this work, we employed a comprehensive infiltration-based modification approach, meaning that DNA probes are present on both the outer and the inner surfaces of the nanochannels. Many previous works conducted systematic studies on the functionalization partitioning of solid-state nanochannels [*ACS Nano* **18**, 7677-7687 (2024); *Nat. Commun.* **9**, 40 (2018); *Nat. Commun.* **9**, 4557 (2018)]. Although the high surface-to-volume ratio within the nanochannels results in the majority of modifications occurring on the inner surfaces of the nanopore membranes, functionalization also takes place on the outer surfaces. Due to the current characterization technology, the representation inside the pore is still very limited. According to the experience of our research team [*Chem. Soc. Rev.* **52**, 6270-6293 (2023)], the modification method of three-step chemical reaction with full infiltration is often used, and the characterization is combined with scanning electron microscope (SEM), contact angle (CA), energy

dispersive X-ray spectroscopy (EDS), X-ray photoelectron spectroscopy (XPS) and *I-V* tests, and it is generally confirmed that the probe has modified the nanoporous channel (**Supplementary Figures 6-8** and **Supplementary Tables 4-10**).

As for reproducibility, each set of experiments was performed in triplicate (or more), and we consistently observed similar results. Additionally, variations in key experimental parameters have been carefully recorded, and multiple batches of functionalized devices have shown comparable performance. The details of our functionalization methods and the reproducibility data are provided in the Supplementary Information to ensure transparency and facilitate a comprehensive understanding of our methodology.

◆ **Reviewer #2 (Remarks to the Author):**

To address the challenge of pH interference in sweat drug detection, the authors propose a nanoconfined biosensor based on a multi-DNA probe co-calibration strategy. This innovative approach combines the high responsiveness of single DNA probes with enhanced signal amplification within a nanoconfined environment, enabling ultra-sensitive drug monitoring across a sweat pH range of 3 to 8. This strategy offers a significantly broader linear response range and enhances sensitivity by 4~5 orders of magnitude. In general, this paper succeeded to show innovation and convincing data analysis. The topic is interesting, and the data is exceptionally plentiful. This paper is well prepared. However, minor revisions are recommended before it can be considered for publication in this journal.

Response: Thank you very much for your comments and suggestions to help us improve the manuscript. We have thoroughly reviewed all the comments and addressed each of your inquiries comprehensively. Furthermore, we have implemented corresponding revisions to both our manuscript and supplementary information. Enclosed at the conclusion of this letter is our detailed response, wherein we have highlighted the modifications made in blue within our revised version. Thank you for viewing.

Comments 1: The design concept of this study is inspired by the distribution of different functional components in biological nanopores. However, this idea is not clearly illustrated in Figure 1. It is suggested to further explain the core concept of the design and provide additional details on how the distribution of different functional components enables efficient multi-DNA probe calibration and enhanced signal recognition. Consider adding schematic diagrams or flowcharts in the figure to more clearly convey this innovative design concept.

Response: Thank you for your insightful feedback regarding the design concept of our study. We have expanded the description in the manuscript to clearly articulate how the distribution of multiple functional components within our solid-state nanopores mimics the arrangement found in biological nanopores. This includes detailing how each DNA probe is strategically positioned to achieve optimal calibration and signal recognition. We have included a more comprehensive explanation of how multi-DNA probe calibration improves signal recognition. This involves describing the interactions between the probes and target molecules, and how these interactions modify the local environment within the nanopore to produce more distinct and reliable electrical signals. These enhancements not only clarify the innovative aspects of our design but also provide a more intuitive understanding of how the multi-DNA probe system functions to improve nanopore-based detection methods.

In this work, inspired by the functions of AG and SF in biological nanopores [*Nat. Commun.* **10**, 5366 (2019); *Small* **18**, 2201925 (2022)], we developed a novel biomimetic solid-state nanochannels functionalized with dual-DNA probes. The biosensor employs cathinone binding aptamer (CBA) as the gate molecule for specific sensing (AG), characterized by its transition from a long-chain structure to a hairpin structure upon binding to the target. Meanwhile, C4 DNA acts as the functional molecule (SF) regulating

selective ions transport, with its structure shifting in response to pH changes, specially forming an i-Motif structure under acidic conditions upon binding with hydron (**Figure R10A**). By co-functionalizing the nanochannels surface with both CBA and C4 DNA (CBA&C4@AAO), co-calibration for target recognition sensitivity and anti-pH interference. Compared to CBA@AAO, the CBA&C4@AAO exhibited a broader response range while maintaining ultra-sensitivity. This enhanced performance is mainly attributed to the reduced hydrophilicity of the nanochannels due to dual aptamer functionalization, which amplifies the current change caused by CBA binding to the target (**Figure R10B**). Additionally, C4 DNA captures hydron from the solution, effectively reducing the signal interference from hydron ion transport. The difference in length between the two DNA probes also increases the density of DNA functionalization on the nanochannel surface, resulting in a more uniformly distributed positive charge. This leads to significant charge changes upon target binding, enhancing the target recognition signal.

Figure R10. (A) In biological nanopores, different functional proteins are distributed, including ion-selective proteins (SF) and specific sensing gate proteins (AG). The cathinone-specific binding probe (CBA) forms a hairpin structure upon binding to cathinone (left diagram). Meanwhile, the pH-responsive C4 DNA undergoes reversible structural changes with varying pH levels (low pH: i-Motif structure, high pH: extended linear structure) (right diagram). **(B)** The dual-DNA probes in the CBA&C4@AAO provide ultra-sensitive target recognition by altering the effective pore diameter, surface wettability, and other nanochannel properties through probe conformational changes, affecting the ionic current signal. The CBA&C4@AAO surface is functionalized with two distinct DNA probes, resulting in a rougher texture compared to the single-probe CBA@AAO, which slightly reduces hydrophilicity and amplifies the effective pore size variation before and after cathinone binding (left diagram). Additionally, the CBA&C4@AAO surface is enriched with C4 DNA, which binds specifically to protons, reducing signal interference caused by proton transmembrane transport (right diagram). The biosensor also shows more pronounced surface charge changes, and this synergistic effect significantly

enhances the biosensor's response to cathinone detection.

We believe that these revisions significantly strengthen the presentation of our design concept and improve the overall clarity and impact of our study. Thank you again for your constructive recommendation, which has been instrumental in enhancing the quality of our manuscript.

Comments 2: The study emphasizes the introduction of multi-DNA probes, which, compared to a single DNA probe, enhances detection sensitivity primarily through changes in the wettability of the nanochannels surface. However, the manuscript lacks detailed experimental data to support this aspect of the research. It is recommended to include relevant experimental data that demonstrate the specific characterization of wettability changes on the nanochannels surface after modification with multi-DNA probes. This will enhance the reliability and persuasiveness of this section of the study.

Response: Thank you for your valuable feedback and for highlighting the importance of characterizing the wettability changes on the nanochannel surfaces. We have conducted additional experiments to specifically evaluate the wettability of the nanochannels after modification with single and multi-DNA probes (**Figure R11**). We performed contact angle (CA) measurements to assess the hydrophilicity and hydrophobicity of the nanochannel surfaces. The results demonstrate that the introduction of multi-DNA probes significantly alters the surface wettability compared to single DNA probe modification. Specifically, the CA increased from $55.1^\circ \pm 0.7^\circ$ for single DNA probes to $62.5^\circ \pm 0.5^\circ \sim 76.5^\circ \pm 0.4^\circ$ for multi-DNA probes, indicating decreased hydrophilicity. This change in wettability contributes to the enhanced detection sensitivity observed in our study by facilitating more effective interactions between the target molecules and the functionalized nanochannels.

Figure R11. Contact angle measurements of DNA probe-functionalized nanopore membranes with varying CBA:C4 DNA ratios.

Additionally, comprehensive characterization results, including the CA measurements and analysis, are provided in the Supplementary Information to offer further evidence supporting our findings. We believe that these additions adequately address your concerns and significantly enhance the reliability and persuasiveness of our study. Thank you again for your constructive recommendation, which has been instrumental in improving the quality of our manuscript.

Comments 3: This work selects two different DNA probes. One is a target-specific recognition aptamer, whose binding interaction with the target has been demonstrated through molecular docking experiments. The other is a pH-specific DNA probe, for which no related theoretical simulation has been conducted, and the data only provides CD spectroscopy. Please provide a detailed explanation of the scientific basis and reliability of the design of these DNA probes.

Response: Thank you for your insightful questions regarding the design and validation of the two different DNA probes used in our work. We appreciate the opportunity to elaborate on the scientific basis and reliability of our probe selection.

The cathinone-binding aptamer (CBA) was meticulously selected for its high specificity and affinity towards the target molecule, cathinone. The binding interaction between CBA and cathinone was initially identified and optimized through molecular docking experiments, which provided a theoretical foundation for understanding the binding mechanics at the molecular level (**Figure 4B**). These docking studies allowed us to predict the binding sites, interaction energies, and conformational changes upon target binding, thereby ensuring that CBA possesses the necessary characteristics for effective target recognition.

Figure R12 (28). Molecular Dynamics Cyclotron Radius Diagram of the Complex (C4 DNA in Different pH Environments). (A) pH = 3; (B) pH = 8. The radius of gyration (R_g) is used to assess the compactness of the structure. A larger R_g value indicates a looser structure, while a smaller value indicates a more tightly packed conformation. The figure shows that after equilibration, the complex exhibits minimal fluctuation, with the system remaining stable at approximately 3.64 nm (pH = 3) and 3.69 nm (pH = 8).

Figure R13 (S29). Average Solvent-Accessible and Total Surface Area of Protein Residues in the Experimental System. (A) pH = 3; (B) pH = 8. Solvent-accessible surface area (SASA) refers to the area of the protein surface that is accessible to solvent molecules, serving as a measure of the protein's surface exposure to the solvent. A lower

SASA value indicates a more compact structure, while a higher value suggests a more extended conformation. As shown in the figure, the solvent-accessible surface area of the entire system remains relatively stable, maintaining values of 73.35 nm² (pH = 3) and 73.71 nm² (pH = 8).

We selected C4 DNA as the pH-specific probe based on its known ability to form pH-responsive structures, such as the i-Motif, under acidic conditions. Our design was informed by literature demonstrating that i-Motif-forming sequences reliably respond to pH changes [*Adv. Funct. Mater.* **25**, 421-426 (2015); *J. Am. Chem. Soc.* **130**, 8345 (2008)]. The i-Motif structure is well-established for its stability and its structural transitions in response to protonation events, making it an ideal candidate for regulating ion transport based on environmental pH variations. Additionally, in this work, we conducted theoretical simulations of C4 DNA in different pH environments (**Figures R12, R13**). Molecular dynamics (MD) simulation results revealed that the Solvent Accessible Surface Area (SASA) of C4 DNA at pH 3 is smaller than at pH 8, indicating that C4 DNA adopts a more compact structure under acidic conditions and a more extended conformation at pH 8. This variation in the SASA metric further supports the ability of C4 DNA to undergo structural modulation in response to pH changes, thereby enhancing its scientific basis and reliability as a pH-specific probe.

To substantiate the functionality of both DNA probes, we employed Circular Dichroism (CD) spectroscopy, which provided critical insights into the conformational changes upon binding interactions. For CBA, CD spectra confirmed the transition from a long-chain to a hairpin structure upon cathinone binding, aligning with our molecular docking predictions (**Figures 4C, D**). For C4 DNA, CD spectroscopy validated the formation of the i-Motif structure under acidic conditions, thereby confirming its pH-responsive behavior (**Figure 4G**).

In summary, the selection of CBA and C4 DNA as multi-DNA probes in our nanopore-based detection system is based on a combination of theoretical insights, empirical validation, and established scientific principles. These comprehensive validation efforts ensure the scientific robustness and reliability of our probe designs, thereby enhancing the overall efficacy and sensitivity of our nanopore-based detection method. We have included additional experimental data and detailed explanations in the revised Supplementary Information to further support the design and reliability of these DNA probes. Thank you again for your valuable feedback, which has enabled us to strengthen our work.

Comments 4: Currently, the XPS characterization data only shows the functionalization of a single DNA probe on the nanopore, and lacks data for the mixed modification of both DNA probes. It is recommended to supplement the XPS data for the co-modification of the two DNA probes and compare it with the results of the single DNA probe. This would not only validate the increase in the surface modification density of the nanopore functionalized with dual aptamers, but also further confirm the effectiveness and advantages of the dual-DNA probe co-calibration strategy.

Response: Thank you for the reviewer's comment. We have already provided

comparative data for the co-modification of both DNA probes would strengthen the validation of our dual-DNA probe co-calibration strategy. In the revised supplementary Information, we have included additional XPS measurements performed on nanopore membranes functionalized with both DNA probes in a mixed ratio (1:1). By comparing the XPS spectra of single-probe-modified and dual-probe-modified nanopores, we can clearly observe changes in the elemental composition and peak intensities corresponding to nitrogen-containing groups (**Figure R14**, and **Supplementary Fig. 7**). The nitrogen content of CBA@AAO NcBs increased by 1.01% before and after functionalization, while that of CBA&C4@AAO increased by 1.46% (**Table R1** and **Supplementary Tables 5-6**). These differences provide direct evidence for the increased probe density and more robust functionalization achieved through the dual-aptamer approach. These new XPS data have been incorporated into the revised manuscript and the Supplementary Information. We believe that this addition not only substantiates the improvements in nanopore functionalization density but also further validates the effectiveness and advantages of the dual-DNA probe co-calibration strategy. Your recommendation has been instrumental in strengthening our presentation and we appreciate your thoughtful input.

Figure R14 (S18). (A) XPS of the surfaces of CBA&C4@AAO (DNA mixing proportion 1:1) nanochannel membrane before and after preparation. (B) Changes of surface N content of CBA@AAO and CBA&C4@AAO NcBs before and after preparation.

Table R1. The XPS data of the CBA&C4@AAO (CBA:C4 DNA = 1:1) before and after preparation.

Name	Start BE	Peak BE	End BE	Height CPS	FWHM eV	Area (P) CPS.eV	At. %
CBA&C4@AAO before preparation							
C1s	297.98	283.71	279.18	37564.77	2.15	101790.92	52.33
N1s	409.98	398.07	392.18	3237.82	2.31	11093.84	3.67
O1s	544.98	530.53	525.18	72611.05	2.67	206928.75	44
CBA&C4@AAO NcBs							
C1s	297.98	283.86	279.18	51809.96	2.28	138769.07	69.27
N1s	409.98	398.08	392.18	4952.13	2.01	15957.73	5.13

Comments 5: The color scheme of the images in the manuscript is generally clear, but some images (such as Figure 4A) require further enhancement of color contrast and clarity to ensure that the details in the image are more prominent and easier to distinguish. It is recommended to adjust the colors accordingly. In addition, the format of the references should comply with requirements of the journal. For example, “*ACS Nano* **18** (8), 6570-6578 (2024).” and “*J. Am. Chem. Soc* **146**, 20, 14058-14066 (2024).” in the references. It is suggested to recheck and ensure that all references are consistent in format. In addition, the details of the manuscript presentation need to be carefully checked, for example, the reference to “supporting information” in the manuscript should be changed to “supplementary information”.

Response: Thank you for your thorough and constructive feedback. We have addressed your comments. We have revisited all figures, including **Figure 4A**, to enhance color contrast and clarity. Adjustments were made to the color schemes to ensure that all details are more prominent and easily distinguishable. These modifications improve the visual quality and make the data presentation clearer for readers. And we have meticulously reviewed and reformatted all references to comply with the journal's guidelines. This ensures consistency and adherence to the required citation style throughout the manuscript. In addition, all instances of “supporting information” in the manuscript have been changed to “supplementary information” to align with the journal's preferred terminology. Furthermore, we conducted a comprehensive review of the manuscript to identify and rectify any other inconsistencies or formatting issues. This includes verifying figure labels and legends to ensure they meet the journal's standards. These revisions have been implemented in the revised manuscript, and the updated figures and references have been incorporated accordingly. We appreciate your valuable suggestions, which have significantly enhanced the quality and clarity of our work. Thank you once again for your insightful comments and support.

We appreciate the Editorial Office's and the reviewers' kind work earnestly. Your careful review has helped to make our study clearer and more comprehensive. Once again, thank you very much for your comments and suggestions.

Response to Comments of the Reviewers

◆ Reviewer #1 (Remarks to the Author):

The authors have done a very thorough job responding to the comments and have updated their manuscript accordingly. I recommend publication without further revision, and this will be a nice addition to Nature Communications.

Response: We are extremely grateful to reviewer 1 for considering that we have fully and satisfactorily addressed all your comments. Your recommendation to publish this work is also highly appreciated.

◆ **Reviewer #2 (Remarks to the Author):**

The authors have fully addressed my previous comments.

Response: We appreciate the reviewer's high comments and recommendation.